# RETHINKING LLM HUMAN SIMULATION: WHEN A GRAPH IS WHAT YOU NEED

## ABSTRACT

Large language models (LLMs) are increasingly used to simulate humans, with applications ranging from survey prediction to decision-making. However, are LLMs strictly necessary, or can smaller, domain-grounded models suffice? We identify a large class of simulation problems in which individuals make choices among discrete options, where a graph neural network (GNN) can match or surpass strong LLM baselines despite being three orders of magnitude smaller. We introduce **G**raph-bas**E**d **M**odels for Human **S**imulation (GEMS), which casts discrete choice simulation tasks as a link prediction problem on graphs, leveraging relational knowledge while incorporating language representations only when needed. Evaluations across three key settings on two simulation datasets show that GEMS achieves comparable or better accuracy than LLMs, with far greater efficiency, interpretability, and transparency, highlighting the promise of graph-based modeling as a lightweight alternative to LLMs for human simulation.

## 1 INTRODUCTION

The use of large language models (LLMs) to simulate human attitudes and behaviors has recently attracted significant attention, driving new subfields of research (Gao et al., 2024; Anthis et al., 2025), conference workshops (SocialSim'25, 2025) and panels (Hwang et al., 2025), and even startups (Expected Parrot, 2025; Artificial Societies, 2025). Recent work has explored LLM human simulation under various names, including generative agents (Vezhnevets et al., 2023; Park et al., 2024), survey prediction (Rothschild et al., 2024; Holtdirk et al., 2025), human simulation (Manning et al., 2024; Wang et al., 2025b; Li et al., 2025; Kolluri et al., 2025; Kang et al., 2025), digital twins (Toubia et al., 2025), pluralistic alignment (Zhao et al., 2024; Feng et al., 2024; Yao et al., 2025), or as foundation models for human cognition and behavior (Binz et al., 2025; Xie et al., 2025).

What ties these efforts together is their common reliance on LLMs. LLMs offer advantages in simulating humans: natural language understanding that supports a wide range of prompts describing context and tasks; broad knowledge of human behavior acquired through large-scale pretraining; and language generation capabilities that span open-ended reasoning and discrete choice. Yet, this raises a central question: is an LLM *always* necessary, or are there settings where simpler, domain-grounded models may suffice if not yield further advantages, such as efficiency, interpretability, and transparency?

**The present work.** We identify a large class of simulation problems where graph neural networks (GNNs)—orders of magnitude smaller than LLMs—match or surpass strong LLM-based methods. While GNNs are not suited for open-ended generation, we show that they either outperform or match performance of LLMs on *discrete choice simulation*, predicting an individual's choice over a set of options given situational context. This class of problems encompasses many popular tasks studied in LLM human simulation literature (Section 3.1).

We formulate discrete choice simulation as a link prediction problem on a graph (Figure 1), with nodes corresponding to individuals and choices, and develop a GNN-based framework for this problem. We refer to our overall approach as **G**raph-bas**E**d **M**odels for Human **S**imulation (GEMS). Unlike prior work that casts the task as next-token prediction in an LLM, GEMS emphasizes learning from relational structures while drawing on language representations *only* when necessary. We evaluate GEMS on three key settings of discrete choice simulation tasks: (1) missing responses (i.e., imputation), (2) new individuals, and (3) new questions. We compare GEMS to a series of

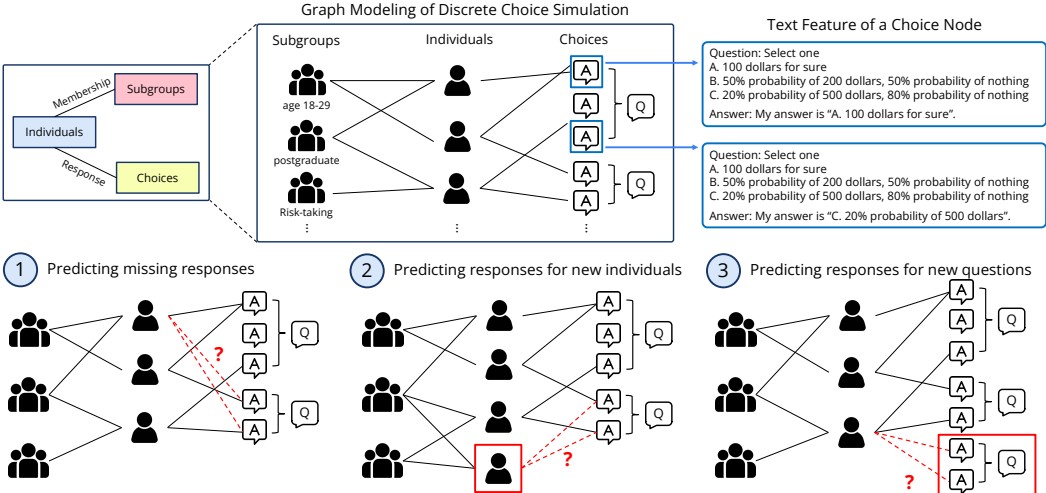

Figure 1: In our GEMS framework, we construct a heterogeneous graph for discrete choice simulation tasks (**Top**) where the goal is to predict the option chosen by an individual human user in response to a context or question. Under three widely-studied sub-settings (**Bottom**), we show that our GNN-based method achieves accuracy comparable to the best LLM-centric approaches.

LLM baselines—zero-shot, few-shot, chain-of-thought prompting, and supervised fine-tuning—on two datasets for human behavior simulation, OPINIONQA (Santurkar et al., 2023) and TWIN-2K (Toubia et al., 2025). In the first two settings, GEMS achieves comparable or better performance, without using *any* language representations. In the final setting where we encounter a new question that necessitates textual information to make any predictions, leveraging a lightweight LLM-to-GNN embedding projection (Sheng et al., 2025) achieves comparable performance.

GNNs that match LLM performance open up new opportunities. First, GEMS has $\sim 10^3$ fewer parameters and up to $10^2$ times less training compute, allowing researchers without as much compute to participate in human simulation research and enabling models to scale to larger datasets. Second, GEMS learns an embedding of each individual and option, enabling us to study similarities between individuals or between options (Figure 4) and understand where the model's predictions come from, as a simple dot product of node embedding pairs. Third, LLMs suffer from opaque pretraining data, leading to contamination risks (Deng et al., 2024) and biases that impair simulation quality (Cheng et al., 2023; Bisbee et al., 2024) while GEMS minimizes these issues by training GNNs from scratch on transparent, domain-specific data.

**Contributions.** Our work makes the following contributions:

**1. Graph-based formulation of human simulation.** We show that a wide class of LLM human simulation tasks can be cast as a link prediction problem on a graph and develop an appropriate GNN architecture for this problem (Section 4).

**2. Competitive performance over three sub-tasks.** We demonstrate that our approach (GEMS) matches strong LLM baselines in three important settings: predicting (1) missing responses (i.e., imputation), (2) responses of new individuals, and (3) responses for new questions (Section 5).

**3. Objectives beyond accuracy.** We illustrate that GNNs offer several advantages over LLMs at comparable performance, including efficiency, interpretability, and transparency (Section 6).

## 2 RELATED WORK

As interest in human simulation has risen sharply in the past few years, LLMs have remained by far the predominant approach, with work often only testing LLM methods and references to this area including "LLM" in its title (e.g., "LLM social simulation" Anthis et al. (2025), "LLM-simulated data" Hwang et al. (2025)). Alongside this growth, work has examined potential pitfalls of LLM-based simulation (Bai et al., 2025; Kapania et al., 2025), including social biases (Cheng et al., 2023) and misportrayals (Wang et al., 2025a). A large class of LLM-based human simulation reduces to

predictions among discrete choices, typically cast as next-token prediction for the LLM. A popular example is predicting survey responses (Santurkar et al., 2023), where the task is to predict the correct token (e.g., 'A') matching the observed human response. Prior work has explored prompting strategies (Dominguez-Olmedo et al., 2023), including few-shot prompting (Hwang et al., 2023) and prompt engineering (Kim & Yang, 2025), as well as conditioning on open-ended narratives (Park et al., 2024; Moon et al., 2024; Rahimzadeh et al., 2025). Recently, fine-tuning has emerged as a promising alternative, either on community-specific text corpora (Chu et al., 2023; He et al., 2024; Li et al., 2024; Feng et al., 2024) or directly on human response data (Cao et al., 2025; Suh et al., 2025; Binz et al., 2025; Xie et al., 2025; Kolluri et al., 2025).

Yet the task remains selecting from a small, fixed set of tokens. Given this finite label space, is language modeling the best approach? We build on this observation and, in $\mathrm{GEMS}$, emphasize the *relational* structure underlying human choices. This approach exploits similar relational dependencies as graph-based recommender systems, where user-item preferences are represented as edges (Ying et al., 2018; Fan et al., 2019; He et al., 2020), but have been absent in the LLM-centric human simulation literature. Ours is the first work to clarify where graph-based modeling can achieve performance comparable to LLMs in human simulation and to provide direct comparisons against LLM baselines. Please see Appendix B for an extended discussion of related work.

## 3 PROBLEM DEFINITION

### 3.1 DISCRETE CHOICE SIMULATION TASKS

We focus on discrete choice simulation tasks, where an individual is presented with a question and a set of options to choose from, and the goal is to predict which option they will choose. This class of problems encompasses several popular simulation tasks, including predicting survey responses, where the question is the survey question and the options are the answer options (Santurkar et al., 2023; Zhao et al., 2024; Feng et al., 2024); social science experiments, where the question is the stimuli and the options are the outcome response scale or labels (Hewitt et al., 2024; Park et al., 2024); game scenarios, where the question is the game setting description and options are the available actions ("give $5 to opponent") (Xie et al., 2024); or voting, where the question asks the individual to vote among candidates or on a proposed policy and the options are the candidates or level of support, respectively (Yu et al., 2024; Kreutner et al., 2025; von der Heyde et al., 2025).

**Terminology.** Given an individual $u$ and a question $q$, with answer options $\mathcal{A}(q)$, our goal is to predict $u$'s response $y_{uq} \in \mathcal{A}(q)$. Each individual has *individual features*: in human simulation tasks, these often include, but are not limited to, demographic variables. We use individual features to define *subgroups*, which are groups of individuals sharing one or more features. We also have *question features* and *option features*. Since we focus on simulation tasks where LLMs have been the dominant approach, these features are often text, i.e., the text of the question and of each option, but our framework is not restricted to text-only features. We define a *choice* as a pair $(q, a)$ of question $q$ and answer option $a \in \mathcal{A}(q)$; its *choice feature* is the concatenation of the question text and option text. We observe a set of prior responses $\mathcal{Y}$, which consists of responses from *seen* individuals (i.e., those with at least one response in $\mathcal{Y}$) and *seen* questions (i.e., those with at least one response in $\mathcal{Y}$). However, we do not observe responses between all pairs of seen individuals and questions.

### 3.2 TASK SETTINGS

We consider three settings of simulating human behavior over discrete choice options widely studied in previous LLM simulation work (Figure 1).

**(1) Missing responses (Imputation).** Given a *seen* individual $u$ with individual features and prior responses in $\mathcal{Y}$, and a *seen* question $q$ with question and option features and prior responses in $\mathcal{Y}$, predict $y_{uq}$, where $y_{uq} \notin \mathcal{Y}$. Prior LLM work studies few-shot prompting and few-shot fine-tuning for this setting (Hwang et al., 2023; Zhao et al., 2024; Kim & Yang, 2025; Kolluri et al., 2025).

**(2) New individuals.** Given a *new* individual $u$, where we observe their individual features but not any prior responses, predict $u$'s responses to seen questions. This setting has been investigated in several simulation works (Santurkar et al., 2023; Moon et al., 2024; Kang et al., 2025; Li et al., 2025) and is also of interest to pluralistic alignment (Feng et al., 2024; Yao et al., 2025) and group response estimation (Suh et al., 2025; Cao et al., 2025).

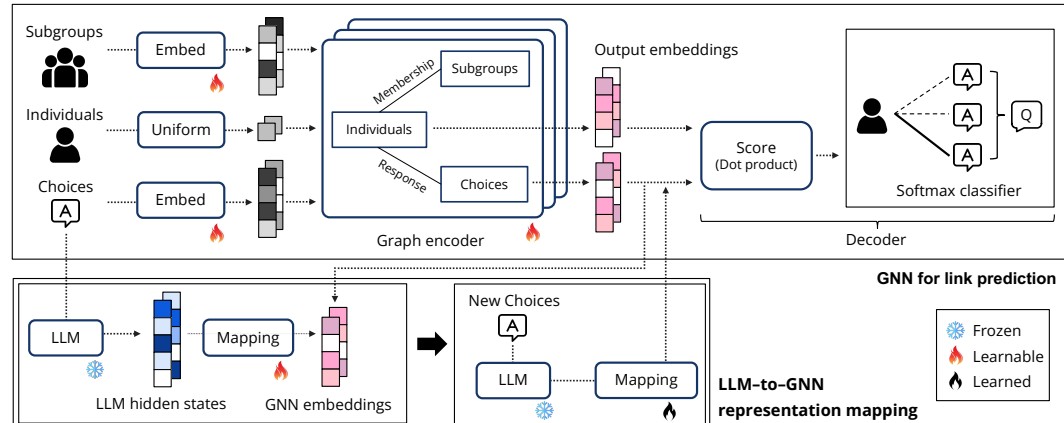

Figure 2: Overall architecture. From the relational structure only, graph encoder learns representations of individual nodes and choice nodes that are subsequently consumed by a dot-product and softmax classifier for response prediction (**Top**). Additionally, from pairs of choice nodes' text feature (LLM hidden states) and GNN output embedding, an LLM-to-GNN representation mapping is learned and used when a new question is presented at setting 3 (**Bottom**).

**(3) New questions.** Given a *new* question $q$, where we observe its question and option features but not any prior responses, predict the responses of seen individuals to $q$. This setting is useful for simulating newly designed items in survey research (Rothschild et al., 2024) or testing generalization to a new simulation setting (Binz et al., 2025; Xie et al., 2025).

## 4 GEMS: GRAPH-BASED MODELS FOR HUMAN SIMULATION

### 4.1 GRAPH REPRESENTATION OF THE TASK

We represent the task as a heterogeneous graph $\mathcal{G}$ with three types of nodes: subgroups $\mathcal{S}$, individuals $\mathcal{U}$, and choices $\mathcal{C}$. Choice nodes are structured as a disjoint union $\mathcal{C} = \mathcal{C}_1 \cup \mathcal{C}_2 \cup \cdots \cup \mathcal{C}_n$, where $\mathcal{C}_q$ is the set of choice nodes for question $q$ and $n$ is the total number of questions. We include two bidirectional relations: membership and response. Membership edges $E_{\mathcal{U}\mathcal{S}}$ with an adjacency matrix $\mathbf{A}_{\mathcal{U}\mathcal{S}} \in \{0,1\}^{|\mathcal{U}| \times |\mathcal{S}|}$ connect each individual to the relevant subgroups. Response edges $E_{\mathcal{U}\mathcal{C}}$ with an adjacency matrix $\mathbf{A}_{\mathcal{U}\mathcal{C}} \in \{0,1\}^{|\mathcal{U}| \times |\mathcal{C}|}$ record which choice an individual chose as a response to a question. Because each question requires selecting one choice, the row-wise sum of $\mathbf{A}_{\mathcal{U}\mathcal{C}}$ is at most $n$.

### 4.2 GNN ARCHITECTURE

Given this graph formulation, we define GEMS as a link prediction model trained end-to-end. As illustrated in Figure 2, an encoder performs relation-aware message passing to produce node embeddings for subgroups, individuals, and choices, and the decoder performs link prediction from the node embeddings. To generalize to *new questions* (Setting 3) whose choice nodes have no edges at test time, we additionally learn an LLM-to-GNN projection that maps choice nodes' text features (frozen LLM hidden states) to representations in the GNN embedding space.

**Input node features.** Individual nodes $\mathcal{U}$ are non-identifiable, thereby assigned a uniform feature $Z_{\mathcal{U}} = \mathbf{1}_{|\mathcal{U}|}$. For subgroup nodes $\mathcal{S}$, we learn input node features via a learnable table $Z_{\mathcal{S}} \in \mathbb{R}^{|\mathcal{S}| \times d_{\mathcal{S}}}$, with a feature dimension $d_{\mathcal{S}}$. For choice nodes $\mathcal{C}$, we also maintain a learnable input feature table $Z_{\mathcal{C}} \in \mathbb{R}^{|\mathcal{C}| \times d_{\mathcal{C}}}$ with a feature dimension $d_{\mathcal{C}}$, not using any textual information.

**Graph encoder.** We adopt standard heterogeneous graph extensions of GNNs, e.g., RGCN, GAT, GraphSAGE (Schlichtkrull et al., 2018; Veličković et al., 2018; Hamilton et al., 2017). Let $z_w^{(0)}$ be the input feature for node $w$ from $Z_{\mathcal{U}}, Z_{\mathcal{S}}$, or $Z_{\mathcal{C}}$. An $L$-layer graph encoder computes

$$z_w^{(\ell+1)} = \sigma\left( \sum_{r \in \mathcal{R}} \left[ \underset{v \in \mathcal{N}_r(w)}{\text{AGG}_r} \phi_r^{(\ell)}\big(z_w^{(\ell)}, z_v^{(\ell)}\big) \right] + \phi_{\text{self}}^{(\ell)}\big(z_w^{(\ell)}\big) \right), \qquad \ell = 0, \ldots, L-1 \quad (1)$$

where $\mathcal{R} = \{\mathcal{U} \to \mathcal{S}, \mathcal{S} \to \mathcal{U}, \mathcal{U} \to \mathcal{C}, \mathcal{C} \to \mathcal{U}\}$ are types of two bidirectional relations (membership and response), $\phi_r^{(\ell)}$ is a relation-specific message passing, $\phi_{\text{self}}^{(\ell)}$ is a self-loop, $\text{AGG}_r$ is a per-relation aggregation over neighbors $\mathcal{N}_r(w)$, and $\sigma$ is a non-linear activation function. We present the details of each function for different GNNs in Appendix E. After the final layer $L$, we apply a node-type-specific linear projection to $z_w^{(L)}$ to obtain the output embedding $z_w^O \in \mathbb{R}^{d_{\text{GNN}}}$ where $d_{\text{GNN}}$ is the dimension of GNN output embeddings.

**Link prediction decoder.** The final GNN decoder consists of a dot-product score function and softmax classifier. For an individual $u \in \mathcal{U}$ and a question $q$ with a set of choice nodes $\mathcal{C}_q \subseteq \mathcal{C}$, the score between the individual and each choice $c \in \mathcal{C}_q$ is obtained as $\text{Dot}(u, c) = (z_u^O)^\top z_c^O$. These scores are then converted to a distribution over choices, with a learnable temperature $\tau$:

$$p(c \,|\, u, q) \;=\; \frac{\exp\big(\text{Dot}(u, c)/\tau\big)}{\sum_{c' \in \mathcal{C}_q} \exp\big(\text{Dot}(u, c')/\tau\big)}. \tag{2}$$

**LLM-to-GNN representation mapping.** In Setting 3 (new questions), the choice nodes for the new question are isolated in the graph since we do not have any responses for that question yet, and have no learned features in the table $Z_{\mathcal{C}}$. Therefore, the graph encoder cannot produce the output embedding for new choice nodes. To make them scorable, we generate a substitute embedding directly from its text features by learning an LLM-to-GNN representation mapping on seen questions. For a choice $c$, the mapping takes a language representation of the choice's text features (a frozen LLM's hidden state $h_{\text{LLM}}(c) \in \mathbb{R}^{d_{\text{LLM}}}$) then outputs $z_c' = \mathbf{W}_{\text{proj}} \, h_{\text{LLM}}(c) \in \mathbb{R}^{d_{\text{GNN}}}$, where $d_{\text{LLM}}$ and $d_{\text{GNN}}$ are dimensions of LLM hidden states and GNN output embeddings, respectively.

The projection is trained on seen choice nodes by matching $z_c'$ to the output node embedding $z_c^O$, inspired by previous work (Sheng et al., 2025; Zhang et al., 2019). At inference for a new question $q$, we compute $z_c'$ for each $c \in \mathcal{C}_q$ and plug these into the decoder in place of $z_c^O$. We note that this mapping is only needed in Setting 3; Settings 1–2 use the output embeddings $z_c^O$ directly.

### 4.3 TRAINING OBJECTIVE

**Link prediction.** Following self-supervised link prediction (Kipf & Welling, 2016; Berg et al., 2017), we train by exposing a subset of train edges to the graph encoder and supervising the model to reconstruct the rest. At each train step we randomly mask response edges from $E_{\mathcal{UC}}$, with a masking strategy defined in Section 5 per setting. For example, say we masked a response edge $(u, c)$ for an individual $u$ and a choice $c$ where $c$ belongs to a question $q(c)$. The decoder generates a probability $p(c \,|\, u, q(c))$ by Equation (2). We aim to minimize the cross-entropy loss

$$\mathcal{L}_{\text{CE}} \;=\; - \sum_{(u,c) \in \text{masked}} \log p(c \,|\, u, q(c)) \tag{3}$$

$\mathcal{L}_{\text{CE}}$ requires no explicit negative sampling: the masked response edge $(u, c)$ is the positive edge, while $(u, c')$ for all $c' \in \mathcal{C}_{q(c)} \setminus \{c\}$ act as implicit negatives through the softmax normalization.

**LLM-to-GNN mapping.** For setting 3 (new question), we learn a linear mapping $\mathbf{W}_{\text{proj}}$ by solving

$$\mathcal{L}_{\text{proj}} \;=\; \sum_{c \in \mathcal{C}_{\text{train}}} \big\| \mathbf{W}_{\text{proj}} \, h_{\text{LLM}}(c) \;-\; z_c^O \big\|_2^2 \;+\; \alpha \, \|\mathbf{W}_{\text{proj}}\|_2^2 \tag{4}$$

where $\mathcal{C}_{\text{train}}$ is the set of choice nodes available during training, $h_{\text{LLM}}$ is a frozen LLM's hidden state for a text feature of a choice node $c$, and $z_c^O$ is the output embedding of an $L$-layer graph encoder. $\alpha$ is a hyperparameter of a ridge regression selected by the prediction accuracy on the validation set.

## 5 EXPERIMENTS

### 5.1 EXPERIMENTAL SETUP

**Datasets.** We evaluate on two simulation datasets: (1) OPINIONQA (Santurkar et al., 2023), comprising responses from 76K individuals to 500 questions spanning various social topics (e.g., political attitudes, media consumption); and (2) TWIN-2K (Toubia et al., 2025), a 150-item battery including economic preferences, cognitive biases, and personality traits, administered to 2K individuals. Examples of questions and choices are provided in Appendix C. Dataset split schemes are described per setting below; graph statistics appear in Appendix D.

**Evaluation metric.** We evaluate performance using accuracy as our metric, comparing the individual's true choice that they selected to the highest-probability choice predicted by the model. Specifically, for a test response edge $(u, c)$ with a question $q(c)$ that $c$ belongs to, the model's prediction is correct if $c = \mathrm{argmax}_{c' \in \mathcal{C}_{q(c)}} p(c'|u, q(c))$ (Equation 2) and incorrect otherwise. Accuracy is the average of correctness over all test response edges.

**Compared methods.** We compare GEMS against five LLM-based baselines (three prompting, two fine-tuning) and include a proxy of lower/upper performance bound. Exact prompt examples for each of the baselines are in Appendix H.

**1.** Zero-shot prompting: Prompt with individual features, following Santurkar et al. (2023).

**2.** Few-shot prompting: Prompt with individual features and the individual's prior responses, following Hwang et al. (2023); Kim & Yang (2025). Not applicable in Setting 2 when no prior responses are available at test.

**3.** Agentic CoT prompting: A chain-of-thought (CoT) framework consisting of a reflection agent and a prediction agent (Park et al., 2024).

**4.** Supervised fine-tuning (SFT): Fine-tune an LLM to predict the answer token given individual features (Cao et al., 2025; Suh et al., 2025; Yao et al., 2025; Kolluri et al., 2025).

**5.** Few-shot fine-tuning (Few-shot FT): Fine-tune an LLM with individual features plus the individual's prior response (Zhao et al., 2024). Like few-shot prompting, not applicable in Setting 2.

**6.** Random (lower bound): Uniformly sample a choice from the question's available options.

**7.** Human retest (upper bound): When available from dataset authors, report test-retest accuracy. It is the probability that the same individual repeats the same choice when re-asked the same question after a fixed time interval (e.g., two weeks).

For main experiments we adopt three language models, LLaMA-2-7B, Mistral-7B-v0.1, Qwen3-8B (Touvron et al., 2023; Jiang et al., 2023; Yang et al., 2025). We also present additional inference results (Dubey et al., 2024; Qwen et al., 2025; OpenAI, 2025) at Appendix F.

## 5.2 SETTING 1: MISSING RESPONSES (IMPUTATION)

Table 1: Accuracy of imputing missing responses. Numbers indicate mean test accuracy with standard deviation from 3 train/val/test random splits with different seeds. $k$ indicates number of in-context examples. For each dataset, bold marks the best accuracy per GEMS and LLM-based methods; underline marks the runner-up. Human retest for OPINIONQA is not available.

| Methods | $k$ | OPINIONQA LLaMA-2-7B | Mistral-7B-v0.1 | Qwen3-8B | TWIN-2K LLaMA-2-7B | Mistral-7B-v0.1 | Qwen3-8B |
|---------|-----|---------|---------|---------|---------|---------|---------|
| Random | | | 27.87 | | | 35.05 | |
| Human retest | | | Not available | | | 81.72 | |
| Zero-shot | | $29.18 \pm 0.15$ | $34.63 \pm 0.19$ | $39.38 \pm 0.20$ | $41.49 \pm 0.31$ | $42.47 \pm 0.27$ | $52.06 \pm 0.38$ |
| Few-shot | 3 | $38.54 \pm 0.21$ | $42.52 \pm 0.06$ | $42.21 \pm 0.08$ | $41.44 \pm 0.88$ | $48.25 \pm 0.73$ | $54.10 \pm 0.51$ |
| | 8 | $37.91 \pm 0.65$ | $45.78 \pm 0.56$ | $43.66 \pm 0.59$ | $43.40 \pm 0.99$ | $51.26 \pm 0.84$ | $56.08 \pm 1.01$ |
| Agentic CoT | 3 | $32.19 \pm 0.25$ | $41.37 \pm 0.47$ | $47.63 \pm 0.17$ | $33.13 \pm 1.57$ | $50.14 \pm 0.93$ | $57.89 \pm 1.80$ |
| | 8 | $28.80 \pm 0.15$ | $38.43 \pm 0.31$ | $47.97 \pm 0.36$ | Context Limit | $48.76 \pm 0.53$ | $60.20 \pm 1.28$ |
| SFT | | $49.41 \pm 0.12$ | $50.56 \pm 0.14$ | $48.84 \pm 0.14$ | $61.23 \pm 0.13$ | $61.85 \pm 0.13$ | $61.49 \pm 0.15$ |
| Few-shot FT | 3 | $55.59 \pm 0.11$ | $\underline{56.31 \pm 0.10}$ | $55.09 \pm 0.14$ | $63.51 \pm 0.15$ | $63.91 \pm 0.16$ | $62.61 \pm 0.19$ |
| | 8 | $55.98 \pm 0.12$ | $\mathbf{56.76 \pm 0.13}$ | $55.61 \pm 0.13$ | $\underline{65.86 \pm 0.17}$ | $\mathbf{66.36 \pm 0.13}$ | $65.27 \pm 0.16$ |
| GEMS | RGCN | | $56.89 \pm 0.12$ | | | $66.36 \pm 0.13$ | |
| | GAT | | $56.40 \pm 0.10$ | | | $66.01 \pm 0.14$ | |
| | SAGE | | $\mathbf{57.00 \pm 0.12}$ | | | $\mathbf{66.62 \pm 0.12}$ | |

**Setup.** We follow the split scheme of Zhao et al. (2024): each dataset is first split at an individual level into 35/5/60% train/validation/test individuals. For each individual held out for validation/test, 40% of their responses are also available during training, while 60% are held out for evaluation. LLM fine-tuning prompts and train graphs are built upon all responses from 35% train individuals and 40% responses from validation/test individuals, having an equal amount of information to train. Validation and test are done on 60% held-out responses for validation/test individuals.

At each training step of GEMS, 50% of response edges in the train graph are randomly masked and used as supervision edges, while all membership edges and unmasked train response edges serve as message passing edges. At validation/test, the entire train graph is used for message passing to predict held-out edges (60% responses from held-out individuals). For LLM few-shot prompts we test 3 or 8 in-context examples, selected from training data by the highest cosine similarity of text embeddings, following Liu et al. (2021); Hwang et al. (2023). Please refer to Appendix E for additional details, e.g., text embedding models and hyperparameter.

**Results.** Table 1 reports test accuracy. GEMS matches or outperforms the strongest LLM-based methods, 8-shot fine-tuning. Performance of LLM-based methods generally improves with more sophisticated prompt design and compute, from zero-shot prompting to few-shot fine-tuning; however, GEMS attains comparable accuracy without using any textual features, relying solely on a learnable feature table over choices and subgroups. We attribute this to the relational structure that alone provides sufficient signal about what the choice has, even in the absence of standalone semantic information. Taken together, these results highlight the value of relational structure for accurate prediction, and make graph modeling with an explicit relational inductive bias a compelling alternative to supervision in the textual modality.

## 5.3 SETTING 2: NEW INDIVIDUALS

Table 2: Accuracy of predicting responses from new, unseen individuals. Numbers indicate mean test accuracy with standard deviation from 3 train/val/test random splits with different seeds. For LLM baselines, few-shot methods are not applicable since we lack any prior responses for new individuals.

| Methods | | LLaMA-2-7B | OPINIONQA Mistral-7B-v0.1 | Qwen3-8B | LLaMA-2-7B | TWIN-2K Mistral-7B-v0.1 | Qwen3-8B |
|---|---|---|---|---|---|---|---|
| Random | | | 27.87 | | | 35.05 | |
| Zero-shot | | $29.15 \pm 0.15$ | $34.40 \pm 0.13$ | $38.97 \pm 0.16$ | $41.57 \pm 0.39$ | $43.03 \pm 0.50$ | $51.79 \pm 0.27$ |
| Agentic CoT | | $18.44 \pm 0.47$ | $33.84 \pm 0.31$ | $39.53 \pm 0.22$ | $21.91 \pm 0.82$ | $45.30 \pm 0.34$ | $53.45 \pm 0.43$ |
| SFT | | $49.35 \pm 0.15$ | $\mathbf{50.49 \pm 0.17}$ | $48.87 \pm 0.16$ | $61.29 \pm 0.22$ | $\mathbf{61.85 \pm 0.19}$ | $61.38 \pm 0.22$ |
| GEMS | RGCN | | $50.50 \pm 0.12$ | | | $62.39 \pm 0.14$ | |
| | GAT | | $50.36 \pm 0.14$ | | | $62.22 \pm 0.14$ | |
| | SAGE | | $\mathbf{50.73 \pm 0.11}$ | | | $\mathbf{62.50 \pm 0.19}$ | |

**Setup.** The split is also done at an individual level: 35% train, 5% validation, and 60% test individuals. In contrast to setting 1 where we hold out 60% responses from each validation/test individual, here we hold out all responses. This disables LLM few-shot prompting at validation and test phases and requires prediction from only individual features. We also modify GEMS training to teach the model how to make predictions for new individuals. At each training step, we randomly select 50% of training individuals, mask all of their response edges to use as supervision edges, and use all membership edges and unselected training individuals' response edges for message passing.

**Results.** Table 2 reports test accuracy. GEMS remains competitive with the strongest LLM-centric baseline, SFT. Trends mirror that of Setting 1: (i) zero-shot prompting and CoT exceed a random baseline and benefit from stronger LLMs but fall behind SFT, and (ii) fine-tuning narrows performance gaps across LLM families. We note that in GEMS, new individuals only have membership edges to subgroup nodes based on their individual features; furthermore, since their input node features are simply $\mathbf{1}_{|\mathcal{U}|}$, their entire predictive signal comes from their subgroup neighbors. By masking out all response edges for 50% individuals during training, we force the learnable subgroup feature $Z_{\mathcal{S}}$ to encode representations that generalize to new individuals, precisely what is needed for the new individual setting. LLM-based methods can acquire similar knowledge by iteratively seeing pairs of individual features and responses, but at a substantially higher computational cost.

## 5.4 SETTING 3: NEW QUESTIONS

**Setup.** We split at the question level into 70/10/20% train/validation/test, following Suh et al. (2025). Validation/test questions are entirely unseen during train; even at test time their choice nodes are isolated in the graph and only text features are available. Responses to train questions from all individuals are used to fine-tune LLMs or to construct the train graph. At validation/test, responses to train questions are reused as in-context examples or as message-passing edges.

Table 3: Accuracy of predicting human responses to new, unseen questions. Numbers indicate mean test accuracy with standard deviation from 3 train/val/test random splits with different seeds. For GEMS, within a row each column indicates the performance with different LLM hidden states. Details about extraction of LLM hidden states can be found at Appendix E.

| Methods | $k$ | OPINIONQA | | | TWIN-2K | | |
| --- | --- | --- | --- | --- | --- | --- | --- |
| | | LLaMA-2-7B | Mistral-7B-v0.1 | Qwen3-8B | LLaMA-2-7B | Mistral-7B-v0.1 | Qwen3-8B |
| Random | | 27.87 | | | 35.05 | | |
| Zero-shot | | $29.15 \pm 0.57$ | $35.60 \pm 2.91$ | $38.84 \pm 1.08$ | $40.03 \pm 2.45$ | $41.30 \pm 3.69$ | $50.94 \pm 2.76$ |
| Few-shot | 3 | $37.93 \pm 2.24$ | $42.49 \pm 3.16$ | $42.74 \pm 2.87$ | $42.09 \pm 3.38$ | $47.88 \pm 1.93$ | $54.02 \pm 4.19$ |
| | 8 | $37.98 \pm 1.62$ | $42.81 \pm 3.39$ | $44.05 \pm 2.65$ | $41.15 \pm 2.77$ | $47.93 \pm 2.30$ | $55.09 \pm 2.50$ |
| Agentic CoT | 3 | $31.46 \pm 2.92$ | $40.20 \pm 1.60$ | $45.90 \pm 3.57$ | $32.16 \pm 3.66$ | $49.67 \pm 3.61$ | $56.18 \pm 2.74$ |
| | 8 | $27.15 \pm 1.42$ | $37.45 \pm 4.94$ | $46.18 \pm 3.70$ | Context Limit | $48.24 \pm 5.43$ | $58.08 \pm 2.83$ |
| SFT | | $44.12 \pm 2.30$ | $47.86 \pm 0.95$ | $43.95 \pm 0.87$ | $55.85 \pm 1.21$ | $56.21 \pm 0.96$ | $56.24 \pm 1.42$ |
| Few-shot FT | 3 | $49.83 \pm 1.53$ | $\underline{51.77} \pm 1.09$ | $49.59 \pm 0.84$ | $58.07 \pm 1.86$ | $59.86 \pm 1.52$ | $59.99 \pm 1.33$ |
| | 8 | $50.11 \pm 1.97$ | $\mathbf{51.83} \pm \mathbf{1.47}$ | $50.00 \pm 1.00$ | $59.87 \pm 1.35$ | $\mathbf{60.84} \pm \mathbf{1.40}$ | $\underline{60.48} \pm 1.79$ |
| GEMS | RGCN | $48.94 \pm 1.71$ | $\mathbf{50.13} \pm \mathbf{1.85}$ | $49.07 \pm 1.48$ | $56.24 \pm 3.65$ | $\mathbf{60.37} \pm \mathbf{2.47}$ | $59.59 \pm 4.42$ |
| | GAT | $46.87 \pm 1.78$ | $49.25 \pm 2.46$ | $48.52 \pm 2.13$ | $52.00 \pm 1.52$ | $56.57 \pm 1.95$ | $57.38 \pm 2.44$ |
| | SAGE | $47.29 \pm 1.89$ | $\underline{49.84} \pm 1.98$ | $49.09 \pm 1.80$ | $54.06 \pm 4.47$ | $58.56 \pm 2.43$ | $\underline{60.03} \pm 3.88$ |

GEMS is trained in two stages. In the first stage, we train the GNN using the link prediction objective (Equation 3) to learn representations of individual and choice nodes in the train graph. To this end, we initially hold out a small fraction (5%) of response edges from the train graph, which we call "transductive validation edges". At each training step, the remaining response edges in the graph are partitioned into 50% supervision edges and 50% message-passing edges. At the checkpoint with the best accuracy on the transductive validation edges, we extract the GNN's output embeddings for choice nodes $\mathcal{C}_{\text{train}}$. In the second stage, we train a projection to map LLM hidden states of $\mathcal{C}_{\text{train}}$ text features to the GNN's output embedding space extracted in stage 1 (Equation 4). At test time, we make predictions for new questions using the projected embeddings of their choices, as described in Section 4.2.

**Results.** Table 3 reports mean accuracy. GEMS equipped with the LLM-to-GNN mapping attains competitive performance. Although GEMS does not outperform fine-tuning LLM (few-shot FT), we show that a lightweight LLM-to-GNN mapping significantly outperforms LLM prompting and closely follows fine-tuning performance. This performance is achieved with LLM hidden states – GNN output embedding pairs from 500 (OPINIONQA) or 150 (TWIN-2K) questions. We also observe that the choice of LLM affects GEMS: accuracies achieved with GEMS correlate with those of the corresponding LLM-based baselines, indicating that gains from stronger LLMs translate through the mapping, consistent with prior observations (Sheng et al., 2025). Taken together, explicitly modeling the relational structure enables encoding meaningful representations of human choices, even generalizable to new questions.

## 6 DISCUSSION ON ADVANTAGES OF GNNS

We have shown that GNNs achieve comparable accuracy to strong LLM baselines on a large class of simulation tasks with discrete choices. This brings several practical advantages: efficiency and scalability, interpretability, and greater transparency, stemming from the relational inductive bias of graphs and the simplicity of our graph encoder-decoder architecture.

**Efficiency and scalability.** As summarized in Figure 3, GEMS attains accuracy comparable to the best LLM-based methods with $\sim 10^2 \times$ less compute and $\sim 10^3 \times$ fewer parameters (see Appendix E.7). GEMS remains tractable as dataset size grows. For example, extrapolating from Figure 3, fine-tuning a single LLM on datasets orders of magnitude larger (e.g., SUBPOP (Suh et al., 2025)) would require on the order of $10^3$ GPU-hours, whereas GEMS trains within a few hours and, under comparable compute, outperforms LLM-based methods.

We attribute this efficiency to the fit between GNNs and the relational structure at the core of discrete choice simulation tasks. Prompt formulations for LLM (Section 5.1) capture at most 1-hop structure and do not naturally express higher-order dependencies such as $u \leftrightarrow c \leftrightarrow u'$ (co-selection of a choice between individuals) or $c \leftrightarrow u \leftrightarrow c'$ (correlated choices mediated by a user). In contrast, GNNs encode these relations explicitly by multi-hop message passing.

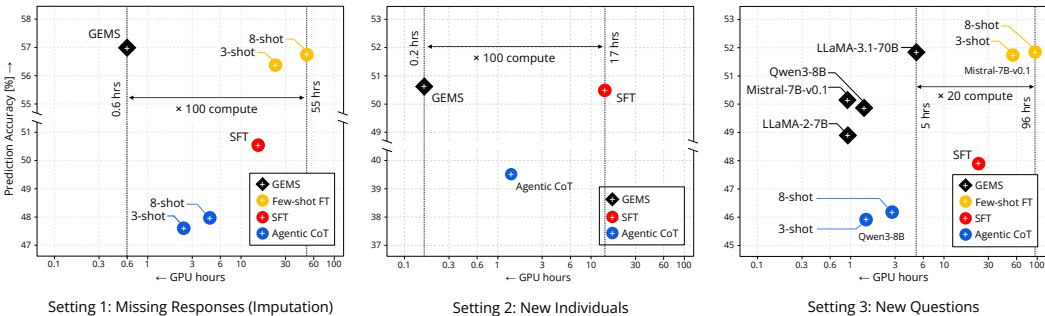

Figure 3: Prediction accuracy *vs.* GPU-hours (A100-80GB) on the OPINIONQA dataset by task setting and method. Zero-/few-shot prompting accuracies fall below the plotted y-range. For LLM-based methods, we report the best result across three LLMs (LLaMA-2-7B, Mistral-7B-v0.1, and Qwen3-8B). For GEMS, we report the best result across three models (RGCN, GAT, and SAGE) for setting 1 & 2, and report across different LLMs for setting 3. See Appendix E for details.

**Interpretability.** GEMS makes predictions in a computationally simple and interpretable way: as a dot product between individual and choice embeddings. In contrast, it is less direct how LLMs combine the description of the individual, question text, and answer options to predict how the individual will respond to the question. Furthermore, we can directly inspect GEMS's embeddings (Figure 4). First, we find that the representations of certain individual features naturally emerge in the embedding space, as the first and second principal components. Second, embeddings of individual nodes show substantial heterogeneity among individuals within the same subgroup, revealing the diversity of individuals beyond their demographics. Third, we find that GEMS encodes nuanced meanings underlying questions and options. In particular, the similarity between two choices with different wordings is reflected as similarity between their embeddings: for example, saying that "reducing illegal immigration" is "a top priority" and "addressing climate change" is "not too important", while the LLM hidden states tend to be overly focused on surface wording similarity (e.g., all "a top priority" are clustered regardless of the topic). Please refer to Appendix F for details.

**Transparency and trust.** LLMs are often trained on undisclosed pretraining data, which creates contamination concerns where evaluation data (e.g., past behavioral studies) may have appeared in the LLMs' training (Deng et al., 2024). Furthermore, LLMs have been shown to display social biases in simulation, such as leaning towards certain groups' opinions (Santurkar et al., 2023), stereotyping (Cheng et al., 2023), or underestimating variance (Bisbee et al., 2024). Finally, prompting-based LLMs are sensitive to prompt format (Lu et al., 2022; Sclar et al., 2024), with many formatting decisions involved in simulation tasks (e.g., order of few-shot examples, format of describing an individual's demographics). All of these issues—opaque training, social biases, and prompt sensitivity—challenge the trustworthiness of LLM-based human simulations. In contrast, GEMS is trained from scratch on task-specific graphs, removing issues of contamination or learning social biases from unknown training data. Furthermore, there is no issue of ordering examples, since the individual is connected to all of their previously selected choices and GNN aggregation is invariant to the order of neighbors (Hamilton et al., 2017). Prompt formatting is only relevant to GEMS when the LLM-to-GNN projection is needed; even then, we find that it exhibits lower variance under prompt perturbations due to the training of the projection matrix.

## 7 CONCLUSION

We introduce GEMS framework to model a large class of LLM human simulation tasks as a link prediction problem on a graph. GEMS learns the relational structure of choices, and uses a lightweight LLM-to-GNN projection only when necessary. Across multiple settings and datasets originally introduced for LLM human simulation, GEMS matches the strongest LLM-based methods. Beyond simulation performance, GEMS offers clear benefits: superior compute and memory efficiency with $\sim 100\times$ less GPU hours and $\sim 1,000\times$ fewer parameters, and a simple decoding that supports inspection and interpretability. Taken together, we suggest: *for human simulation tasks on discrete choices, a graph is what you need.*

## 8 USE OF LLM AND REPRODUCIBILITY STATEMENT

Following the submission guidelines, we note that we used generative AI (ChatGPT) to (1) help locate related work and relevant domains, (2) find potential bugs in the experiment code implementations, and (3) edit writing for potential grammatical mistakes. All ideation, methodological design, experiments, and analyses are conducted by the authors.

We document implementation details for GEMS and all LLM baselines in Appendix E and data preprocessing in Appendix D. We release the training code in an anonymized repository[1]. Because the experiments use individual-level data governed by data use agreements with the original curators, we do not redistribute the raw datasets. Upon acceptance, we will provide the full pipeline, including step-by-step instructions for obtaining access to the data from the providers and the preprocessing code.

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

## A  Limitations, Potential Risks, and Ethical Considerations

### A.1  Limitations

**Limitations of the graph construction.** In Figure 1, we encode individual features via subgroup nodes and connect individual nodes to subgroup nodes with membership edges. The formulation is flexible: it admits different subgroup granularities (e.g., intersectional groups), alternative features (e.g., psychometrics test results), or even peer–to–peer topology that links individuals by social ties as in social recommendation (Fan et al., 2019). However, in the datasets used here, only demographic attributes were available as individual features. Exploring alternative graph constructions with richer features and analyzing their effects is an important future work.

**Limitations of dataset coverage.** Experiments use OPINIONQA and TWIN-2K, both U.S.-centric datasets whose questions and options reflect the design principle of dataset curators. Generalization to other countries or languages is untested. We note, however, that GEMS primarily learns from relational structure and uses language representations only when necessary (Setting 3), making it less sensitive to the interface language than LLM-based simulation methods. By contrast, prior works document that LLM performance can vary substantially by language; this has been shown in public opinion simulation across countries (Qu & Wang, 2024) as well as in multilingual benchmarks (Singh et al., 2024). Accordingly, GEMS may offer robustness when linguistic variation is large, though this claim should be validated empirically with non-English contexts.

**Limitations of performance comparisons.** Our LLM-based methods are fine-tuned up to ∼10B parameters. Larger models may further improve with fine-tuning. However, our experimental results show that after SFT or few-shot fine-tuning, performance gaps across LLMs narrow (Table 1, 2, 3), indicating that GEMS would remain competitive to fine-tuning larger LLMs. Also, our compute figures (GPU hours, parameter counts) are not definitive in the sense that they vary with hardware, quantization, implementation of kernels, and more. To support more informed comparison, we present the implementation details in Appendix E.

**Interpretability claims.** Dot-product decoding based on output node embeddings makes the mechanics of prediction transparent (scores factor as similarities), but they are not causal explanations. Qualitative inspection may risk being misread as normative judgments about groups. Therefore, we suggest using them as diagnostic tools, complemented by ablations and sensitivity analyses, rather than as causal accounts.

### A.2  Potential Risks and Ethical Considerations

**Privacy.** The graph in Figure 1 is constructed from de-identified individual features and response histories provided under the original data providers' terms of use (PewResearch, 2018; Toubia et al., 2025). We neither collect nor store direct identifiers (e.g., names, addresses, phone numbers), and all analyses are performed on anonymized records. To reduce identification risk, we report aggregate metrics (e.g., mean test accuracy) and do not release person-identifiable outputs. For future work, we recommend treating individual-level data as sensitive, especially when it may include personal identifiers or pertains to high-stakes domains (e.g., health), and adhering to applicable regulations, institutional review, and data-security best practices.

**Misuse for high-stakes decisions.** Even when prediction accuracy is high, simulated responses must not replace human participants for decisions affecting rights or access (e.g., hiring, credit, medical triage). Use is inappropriate for targeted surveillance or differential treatment of protected classes. Encoding people via demographic membership edges can inadvertently reinforce stereotypes or obscure within-group heterogeneity; over-reliance on subgroup signals risks reproducing historical biases rather than revealing true relations. Therefore, we are against any deployment without governance, informed consent, and human oversight aligned with ethical guidelines.

## B  Extended Related work

We continue the discussion of related work in Section 2.

**Graph-based recommenders and GNNs.** Relational inductive biases are central to graph recommenders that represent user–item interactions as edges (Battaglia et al., 2018). From GC-MC (Berg et al., 2017), GNNs explicitly leverage higher-order connectivity, including PinSage (Ying et al., 2018), NGCF (Wang et al., 2019b), and simplified designs like LightGCN (He et al., 2020); knowledge-graph–aware models capture attribute/item relations (Wang et al., 2019a); complementary directions capture session and social structures (Wu et al., 2019; Fan et al., 2019) or harness contrastive signals on graphs (Wu et al., 2021; Yu et al., 2023). These successes suggest that human attitudes and behaviors are inherently relational. GEMS draws on these insights but targets a domain currently dominated by LLM: text-based discrete choice human simulations.

**Text-attributed graphs (TAGs).** TAGs integrate node and relation's text attributes with graph topology, letting models enjoy complementary signals. Early work injected text features into matrix-factorization formulations or constructed word–document graphs (Yang et al., 2015; Yao et al., 2019). More recently, an LLM-to-GNN interplay has emerged: (i) LLM as encoder/feature generator, using an LLM as an encoder whose embeddings serve as GNN inputs (Zhu et al., 2021); (ii) alternating, EM-style training that decouples text and graph modules while co-training them via variational objectives (Zhao et al., 2022); and (iii) prompting LLMs to generate descriptions or explanations that enrich node attributes (He et al., 2023). A complementary line of work conditions LLMs on graph structure through prompting and in-context learning, including PRODIGY (Huang et al., 2023), AskGNN (Hu et al., 2024), and GraphICL (Sun et al., 2025). Related efforts project graphs directly into an LLM's token space or align GNN embeddings with token embeddings so that an LLM can reason over graph tokens (Tang et al., 2024; Chen et al., 2024; Wang et al., 2024). In recommender systems, the inverse mapping uses LLM representations within learned graph / collaborative-filtering spaces or co-trains them with GNNs (Ren et al., 2024; Sheng et al., 2025). Collectively, these works underscore the complementarity of language and graph signals. GEMS leverages these insights for human simulation on discrete choice tasks, emphasizing relational structure while drawing on language representations when they are strictly necessary.

## C DATASET DETAILS

### C.1 OPINIONQA

OPINIONQA (Santurkar et al., 2023) is a curated subset of the American Trends Panel (ATP) (PewResearch, 2018). It comprises 500 contentious questions drawn from 14 ATP survey waves, selected for large inter-group response differences. For each anonymized participant, information across 9 demographic traits (age, gender, race or ethnicity, highest level of education, annual income, Census Bureau regions, religion, political affiliation, and political ideology) and their response to survey questions are available. Survey items span a wide range of social topics, including race, politics, age-specific attitudes, media consumption, and views on the future of AI. Owing to its breadth and diversity, OPINIONQA has become a popular dataset for LLM-based human simulation or pluralistic preference alignment research (Hwang et al., 2023; Kim & Yang, 2025; Feng et al., 2024; Zhao et al., 2024; Moon et al., 2024; Suh et al., 2025; Kolluri et al., 2025). Here we present three example questions from the dataset.

---

**Question Example: OpinionQA (1)**

Question: Which of the following would you say you prefer for getting news?

A. A print newspaper
B. Radio
C. Television
D. A social media site (such as Facebook, Twitter or Snapchat)
E. A news website or app

---

---

**Question Example: OpinionQA (2)**

Question: In the future, what kind of an impact do you think the military will have in solving the biggest problems facing the country?

A. A very positive impact
B. A somewhat positive impact
C. A somewhat negative impact
D. A very negative impact

---

**Question Example: OpinionQA (3)**

Question: For each, please indicate if you, personally, think it is acceptable. Casting an actor to play a character of a race or ethnicity other than their own

A. Always acceptable
B. Sometimes acceptable
C. Rarely acceptable
D. Never acceptable
E. Not sure

---

## C.2 TWIN-2K

Twin-2K (Toubia et al., 2025) is a four-wave, nationally representative U.S. panel fielded in January – February 2025 on Prolific for LLM human simulation. Each participant completed questions spanning demographic information, personality scales, cognitive ability tests, economic preference, heuristics-and-biases experiments, etc. Among all questions from Twin-2K, we filtered for multiple-choice questions by removing short answer questions, resulting in 150 questions total. The authors release the full dataset publicly to support broader social-science research.

---

**Question Example: Twin-2K (1)**

Choose an option.

A. I don't feel like a failure
B. I feel that I have failed more than the average person
C. As I look back on my life, all I can see is a lot of failures
D. I feel I am a complete failure as a person

---

**Question Example: Twin-2K (2)**

You have recently graduated from university, obtained a good job, and are buying a new car. A newly designed seatbelt has just become available that would save the lives of 95% of the 500 drivers a year who are involved in a type of head-on-collision. (Approximately half of these fatalities involve drivers who were not at fault.) The newly designed seatbelt is not yet standard on most car models. However, it is available as a $500 option for the car model that you are ordering. How likely is it that you would order your new car with this optional seatbelt?

A. very unlikely
B. unlikely
C. somewhat unlikely
D. somewhat likely
E. likely
F. very likely

---

> **Question Example: Twin-2K (3)**
>
> Antonym: Select the word that is most nearly the opposite in meaning to DEARTH
>
> A. birth
> B. brevity
> C. abundance
> D. splendor
> E. renaissance

## D GRAPH STATISTICS

### D.1 OPINIONQA

We followed the dataset filtering process of Zhao et al. (2024). Beginning with 76K participants in OPINIONQA dataset, filtering to those who answered at least 30 questions yields 19K individuals, 284 survey questions, and 695K (individual, question, choice) triples. 284 survey questions have total 1,103 choices, indicating that each survey question has 3.88 available choices on average. As can be seen from the number of individual nodes 19K much larger than the number of choice nodes 1,103, choice nodes have an order of large node degree compared to individual nodes.

To define subgroup nodes (Figure 1), we employ the 9 demographic attributes used in previous works (Santurkar et al., 2023): age, gender, race or ethnicity, highest level of education, annual income, Census regions, religion, political affiliation, and political ideology. This results in 48 subgroup nodes as follows:

**Age** : 18-29, 30-49, 50-64, 65+

**Race or ethnicity** : White, Black, Hispanic, Asian, Other

**Gender** : Male, Female, Other

**Education** : Less than high school, High school graduate, Some college, no degree, Associate's degree, College graduate / some postgrad, Postgraduate

**Annual income** : Less than $30,000, $30,000–$50,000, $50,000–$75,000, $75,000–$100,000, $100,000 or more

**Region** : Northeast, Midwest, South, West

**Religion** : Protestant, Roman Catholic, Mormon, Orthodox, Jewish, Muslim, Buddhist, Hindu, Atheist, Agnostic, Other, Nothing in particular

**Political affiliation** : Republican, Democrat, Independent, Something else

**Political ideology** : Very conservative, Conservative, Moderate, Liberal, Very liberal

Since the number of individual nodes (19K) is much larger than the number of choice nodes (1,103), choice nodes have a much higher average degree.

We note that there can be different constructions of subgroup nodes, either by considering additional individual features (e.g., marital status, risk-taking preference, etc.) or intersectional attributes as a single subgroup node (e.g., construct a subgroup node representing 'age 18-29 male'). Future work can design their own subgroup nodes tailored to the specific need, and our construction is easily generalizable in those settings.

### D.2 TWIN-2K

TWIN-2K dataset includes both multiple choice and short answer questions. To align with our focus on discrete choice human simulation tasks, we exclude short-answer items, yielding 150 multiple-choice questions. Because nearly all of the 2,000 participants responded to most multiple choice questions, no individuals were removed by the minimum-30-responses criterion. The dataset authors (Toubia et al., 2025) collect demographics using the same categories as Santurkar et al. (2023): we reuse the identical 48 subgroup definitions as in OPINIONQA. The resulting graph contains 48 subgroup nodes, 2,000 individual nodes, 539 choice nodes, and 297K response edges.

# E  IMPLEMENTATION DETAILS

This section details the implementation of the GNN (Section 4.2) and the LLM baselines (Section 5.1). We first present the general GNN training configuration in Appendix E.1, followed by the learnable input embedding tables in Appendix E.2. Next, we instantiate the generic encoder in equation 1 with three architectures—RGCN, GAT, and GraphSAGE—in Appendices E.3 to E.5, respectively. Appendix E.6 describes the setup for the LLM-based methods. Finally, Appendix E.7 compares the model size of LLMs and GEMS GNNs.

## E.1  GNN TRAINING CONFIGURATION

We implement GNNs based on PyTorch Geometric (Fey & Lenssen, 2019). All trainable components of the GNN (learnable input embedding tables, graph encoder, and decoding temperature) are optimized with AdamW optimizer (Loshchilov & Hutter, 2017) using a learning rate of $5 \times 10^{-4}$ and weight decay of $10^{-3}$. We use a cosine annealing learning rate scheduler and apply gradient clipping with a max norm of $0.1$.

Each GNN is trained for 500 epochs with a patience of 20 (i.e., how many epochs the model would continue training after the validation loss stopped from decreasing). In Section 6, the reported GNN training time is measured from the beginning of the training until the termination by exceeding patience. An epoch consists of $50n$ steps, where $n$ is the number of training graphs. Concretely, each training graph is sampled 50 times per epoch with independently re-drawn edge masks that split train response edges into message-passing edges and supervision edges. This resampling reduces overfitting to a fixed edge partition and consistently improves validation accuracy.

**Setting 3 (Predicting Responses for New questions).**   After the GNN is fully trained, we learn an LLM-to-GNN mapping as described in equation 4. The mapping is obtained by solving ridge regression with regularization strength $\alpha$. Rather than a cross-validation, we choose $\alpha$ by directly maximizing validation prediction accuracy on held-out validation questions. In Section 6, the training time of LLM-to-GNN mapping is calculated as the time to extract LLM hidden states from textual features of choice nodes, since solving the ridge regression takes negligible amount of time. In practice, $\alpha \in [50, 800]$ performs best.

## E.2  LEARNABLE INPUT FEATURE TABLE

In Section 4.2, we denote a learnable input feature table for subgroup nodes $\mathcal{S}$ as $Z_{\mathcal{S}} \in \mathbb{R}^{|\mathcal{S}| \times d_{\mathcal{S}}}$ and choice nodes $\mathcal{C}$ as $Z_{\mathcal{C}} \in \mathbb{R}^{|\mathcal{C}| \times d_{\mathcal{C}}}$. We set $d_{\mathcal{S}} = 16$ and $d_{\mathcal{C}} = 128$ for all settings on the OPINIONQA dataset, and $d_{\mathcal{S}} = 8$ and $d_{\mathcal{C}} = 64$ for all settings on the Twin-2K dataset.

## E.3  RELATIONAL GRAPH CONVOLUTION (RGCN)

We use a 2-layer RGCN (Schlichtkrull et al., 2018). Following the feature table dimension in E.2, input dimensions are $(16, 1, 128)$ for (subgroup, individual, choice) nodes on OPINIONQA dataset and $(8, 1, 64)$ on Twin-2K dataset. All hidden layers use the choice node's input dimension, i.e., 128 for OPINIONQA and 64 for Twin-2K.

In equation 1, we present the general graph encoder forward pass as

$$z_w^{(\ell+1)} = \sigma\!\left( \sum_{r \in \mathcal{R}} \left[ \operatorname*{AGG}_{\substack{r \\ v \in \mathcal{N}_r(w)}} \phi_r^{(\ell)}\!\big(z_w^{(\ell)}, z_v^{(\ell)}\big) \right] + \phi_{\text{self}}^{(\ell)}\!\big(z_w^{(\ell)}\big) \right), \qquad \ell = 0, \dots, L-1 \quad (5)$$

For RGCN, we use ReLU as the non-linear activation $\sigma$ and a mean pooling for $\operatorname{AGG}_r$ for all relations $r$. Following the standard RGCN implementation, a relation-specific message passing is

$$\phi_r^{(\ell)}\!\big(z_w^{(\ell)}, z_v^{(\ell)}\big) = \frac{1}{|\mathcal{N}_r(w)|} \mathbf{W}_r^{(\ell)} z_v^{(\ell)}, \tag{6}$$

where the learnable $\mathbf{W}_r^{(\ell)}$ maps from the layer-$\ell$ embedding of the node $v$ to the layer-$(\ell+1)$ embedding dimension of node $w$; the factor $|\mathcal{N}_r(w)|^{-1}$ provides degree normalization for relation $r$.

Similarly, self-loops use a learnable matrix $\mathbf{W}_{\text{self}}^{(\ell)}$

$$\phi_{\text{self}}^{(\ell)}\big(z_w^{(\ell)}\big) = \mathbf{W}_{\text{self}}^{(\ell)} z_w^{(\ell)}. \tag{7}$$

Additionally, we apply post-activation LayerNorm (Ba et al., 2016) and dropout with rate $0.5$ at all layers of the graph encoder.

### E.4 GRAPH ATTENTION NETWORK (GAT)

The equation 1 is implemented with a multi-head Graph Attention Network (Veličković et al., 2018) as

$$z_w^{(\ell+1)} = \sigma\left(\sum_{r\in\mathcal{R}}\left[\,\|_{h=1}^{H_\ell}\sum_{v\in\mathcal{N}_r(w)\cup\{w\}}\alpha_{wv,r}^{(\ell,h)}\,\boldsymbol{\Theta}_{t,r}^{(\ell,h)}\,z_v^{(\ell)}\right]\right), \qquad \ell = 0,\dots,L-1, \tag{8}$$

where $\|$ denotes concatenation across heads, $h$ indicates the head index ranging from 1 to the number of heads in the $\ell$-th layer ($H_\ell$), and the attention coefficient $\alpha$ for the layer-$\ell$ head-$h$ relation-$r$ from the source node $v$ to the target node $w$ is

$$\alpha_{wv,r}^{(\ell,h)} = \operatorname*{softmax}_{v\in\mathcal{N}_r(w)\cup\{w\}}\Big(\operatorname{LeakyReLU}\Big(\mathbf{a}_{s,r}^{(\ell,h)\top}\boldsymbol{\Theta}_{s,r}^{(\ell,h)}z_w^{(\ell)} + \mathbf{a}_{t,r}^{(\ell,h)\top}\boldsymbol{\Theta}_{t,r}^{(\ell,h)}z_v^{(\ell)}\Big)\Big) \tag{9}$$

where $\mathbf{a}_{s,r}^{(\ell,h)}$ and $\mathbf{a}_{t,r}^{(\ell,h)}$ are learnable source and target scoring vectors, $\boldsymbol{\Theta}_{s,r}^{(\ell,h)}$ and $\boldsymbol{\Theta}_{t,r}^{(\ell,h)}$ are learnable source and target feature transformation matrix, and $\operatorname{LeakyReLU}$ is a LeakyReLU function with a negative slope of $0.2$ as in the default implementation of PyTorch Geometric. Softmax is performed over all neighboring nodes of $w$ defined by the relation $r$ and $w$ itself.

We use a 2-layer GAT. Following the input table dimension in E.2, input feature dimensions are $(16, 1, 128)$ for (subgroup, individual, choice) nodes on the OPINIONQA dataset and $(8, 1, 64)$ on Twin-2K. All hidden layers use the choice node's input dimension (128 for OpinionQA, 64 for Twin-2K) with 4 heads in the first layer (per-head size 32) and 1 head in the second layer (per-head size 128), keeping the hidden size unchanged across layers.

We set $\sigma = \operatorname{ReLU}$, and apply post-activation LayerNorm (Ba et al., 2016). We also apply dropout at rate $0.4$ to the normalized attention coefficients $\alpha$ and at rate $0.5$ to the post-activation node embeddings between layers.

### E.5 GRAPHSAGE

We instantiate the generic graph encoder in equation 1 with a GraphSAGE operator (Hamilton et al., 2017). For each relation $r \in \mathcal{R}$ and given a target node $w$, we first compute a relation-specific mean-pooled neighbor message

$$m_{w,r}^{(\ell)} = \operatorname*{MEAN}_{v\in\mathcal{N}_r(w)}\big(\boldsymbol{\Theta}_r^{(\ell)}\,z_v^{(\ell)}\big), \tag{10}$$

where $\boldsymbol{\Theta}_r^{(\ell)}$ is a learnable matrix that maps the layer-$\ell$ embedding of a source node $v$ to the layer-$(\ell+1)$ embedding space of the target node for relation $r$. Messages from all relations are summed and combined with a learnable root (self) transformation $\boldsymbol{\Theta}_{\text{self}}^{(\ell)}$. Subsequently, the embedding is $L2-$normalized and passed through a non-linear activation:

$$z_w^{(\ell+1)} = \sigma\left(\operatorname{Normalize}\left(\boldsymbol{\Theta}_{\text{root}}^{(\ell)}z_w^{(\ell)} + \sum_{r\in\mathcal{R}}m_{w,r}^{(\ell)}\right)\right), \tag{11}$$

We set $\sigma = \operatorname{ReLU}$, apply post-activation LayerNorm (Ba et al., 2016) at every layer, and use dropout with rate $0.5$ on the post-activation node embeddings between layers.

We use a 2-layer GraphSAGE. Following the input feature dimensions in Appendix E.2, input sizes are $(16, 1, 128)$ for (subgroup, individual, choice) nodes on OPINIONQA and $(8, 1, 64)$ on Twin-2K. All hidden layers use the choice-node width, i.e., 128 for OPINIONQA and 64 for Twin-2K.

### E.6 LLM

For all LLM prompting experiments, we used 2×NVIDIA A100 80GB (SXM4) and vLLM framework (Kwon et al., 2023). For selecting in-context examples in few-shot prompting and Agentic CoT, we encode each question text with `gemini-embedding-001` embedding model and compute cosine similarities between the target question and candidate in-context example questions. Following Hwang et al. (2023), the selected examples are ordered by ascending cosine similarity, from least to most similar. To ensure consistent information access across methods, in-context examples are drawn exclusively from the training set and not from the validation set (Sections 5.2, 5.4).

For all LLM fine-tuning methods, we used 4×NVIDIA A100 80GB (SXM4) and built on Llama-cookbook codebase. Each run trained for three epochs using the model's default precision, and we selected the checkpoint with the lowest validation loss. We largely followed hyperparameter setting of Suh et al. (2025), tuning the learning rate over {1e-4, 2e-4, 4e-4} and settled on 2e-4. Training used LoRA (Hu et al., 2022) with rank 8, $\alpha = 32$, and dropout 0.05, applied to the attention query and value matrices. We optimized with Adam optimizer (Kingma & Ba, 2014) and the effective batch size was fixed to 256 by setting per-GPU batches and gradient accumulation steps to fit memory.

### E.7 MODEL PARAMETERS

In this section, we report parameter counts for GEMS and the LLMs, following the implementation details in the previous sections. For LLMs fine-tuned with LoRA, the trainable parameter count equals the number of LoRA adapter parameters, much smaller than the total parameter count. Because both training and inference still require loading the full model, we use total parameter count when comparing the model size. The size of the GNN (GEMS) varies between datasets because we select the hidden dimension per dataset, as noted above.

Table 4: Number of parameters for each model. K, M, and B stand for $10^3$, $10^6$, $10^9$, respectively.

| # parameters | LLM | | | GEMS (RGCN) | |
|---|---|---|---|---|---|
| Model | LLaMA-2-7B | Mistral-7B-v0.1 | Qwen3-8B | - | |
| Dataset | | - | | OPINIONQA | TWIN-2K |
| Trainable | 4.19 M | 3.41 M | 3.83 M | 420 K | 111 K |
| Total | 6.61 B | 7.24 B | 8.19 B | 420 K | 111 K |

## F ADDITIONAL EXPERIMENT RESULTS

### F.1 EMBEDDING VISUALIZATION

Figure 4 visualizes LLM hidden states and GNN embeddings for four example questions in OPINIONQA dataset. Each question asks how high a priority the federal government should give to an issue: (B) reducing illegal immigration, (C) reducing economic inequality, (D) addressing climate change, and (F) reducing gun violence, with four response options ranging from 'top priority' to 'should not be done.' This results in 16 choice nodes in total. All plots show the first two principal components of principal component analysis (PCA).

**Embedding structure of choice nodes.** The top left panel plots the LLM hidden states for the 16 choice nodes: points cluster by option, producing four clusters one per option but not clearly indicating what semantic meaning each choice has. The top right panel shows choice nodes' output node embeddings after the first training stage described in Section 5.4. Here, choices for three questions (C, D, F) are located along a common one-dimensional trajectory in the PCA plane, whereas the choices for question B align along a distinct trajectory. From this observation, we can infer that three questions (C, D, F) are closely related while one question (B) sits on a slightly different social issue dimension, which is consistent with prior observation from survey researchers (Center, 2024).

**Embedding structure of individual nodes.** The remaining panels plot GNN output node embeddings of individual nodes, with colors indicating an individual feature per panel (annual income, political ideology, age, or gender). The PCA axes exhibit interpretable variation: PC1 aligns most strongly with political ideology feature and PC2 with income. Yet, points within any given subgroup remain dispersed, indicating substantial within-group heterogeneity. We note that the prediction is made by taking dot-product between each individual node embedding and the four choice node embeddings, followed by the softmax for multinomial distribution over options.

## F.2 PREDICTION WITH DIFFERENT LLMS

In this section, we report LLM inference results across multiple models. We expand the evaluation to additional LLMs on the predicting missing responses setting of the OPINIONQA dataset to examine performance differences by model family and size. Consistent with Tables 1 to 3, larger and more recent models generally perform better, with the largest gains appearing under the Agentic-CoT method where reasoning ability is most critical. This trend is most pronounced for Qwen3, a reasoning model family. While we did not conduct extensive fine-tuning (SFT or few-shot fine-tuning) on larger models due to computational constraints (Section 6), we hypothesize that GEMS would remain competitive with fine-tuned large LLMs, as fine-tuning tends to compress between-model variance in accuracy (Section 5).

Table 5: An extended evaluation of LLMs. Here we report the values on a setting 1 (missing responses) to compare the performance differences of LLMs before fine-tuning.

| Methods Release date | k | LLaMA-2-7B July 2023 | LLaMA-2-70B July 2023 | LLaMA-3.1-8B July 2024 | LLaMA-3.1-70B July 2024 | Mistral-7B-v0.1 Sep. 2023 | Mistral-Small-24B-2501 Jan. 2025 | Qwen2.5-7B Sep. 2024 | Qwen3-8B Apr. 2025 | Qwen3-32B Apr. 2025 | GPT-OSS-20B Aug. 2025 |
|---|---|---|---|---|---|---|---|---|---|---|---|
| **Setting 1 (Missing responses), OPINIONQA** | | | | | | | | | | | |
| Random | | | | | | | 27.87 | | | | |
| Zero-shot | | 29.15 | 36.47 | 38.71 | 43.45 | 35.60 | 41.79 | 40.07 | 38.84 | 40.42 | 35.71 |
| Few-shot | 3 | 37.93 | 40.76 | 44.04 | 46.04 | 42.49 | 45.85 | 42.56 | 42.74 | 43.53 | 42.18 |
| | 5 | 39.41 | 43.26 | 44.22 | 47.35 | 44.43 | 45.82 | 41.40 | 42.69 | 46.04 | 44.53 |
| | 8 | 37.98 | 40.34 | 42.26 | 44.55 | 42.81 | 45.27 | 42.53 | 44.05 | 44.28 | 41.47 |
| | 13 | 37.78 | 42.45 | 43.24 | 49.66 | 44.03 | 46.41 | 42.49 | 44.03 | 46.72 | 42.50 |
| Agentic CoT | 3 | 32.19 | 43.66 | 42.80 | 49.32 | 41.37 | 46.01 | 43.82 | 47.63 | 47.57 | 44.42 |
| | 5 | 30.72 | 45.82 | 42.96 | 49.55 | 38.92 | 48.22 | 43.58 | 47.92 | 48.40 | 45.90 |
| | 8 | 28.80 | 42.63 | 42.15 | 47.72 | 38.43 | 46.24 | 41.04 | 47.97 | 48.44 | 41.94 |
| | 13 | 28.05 | 45.34 | 42.81 | 48.06 | 38.37 | 48.67 | 39.83 | 48.88 | 50.70 | 44.03 |

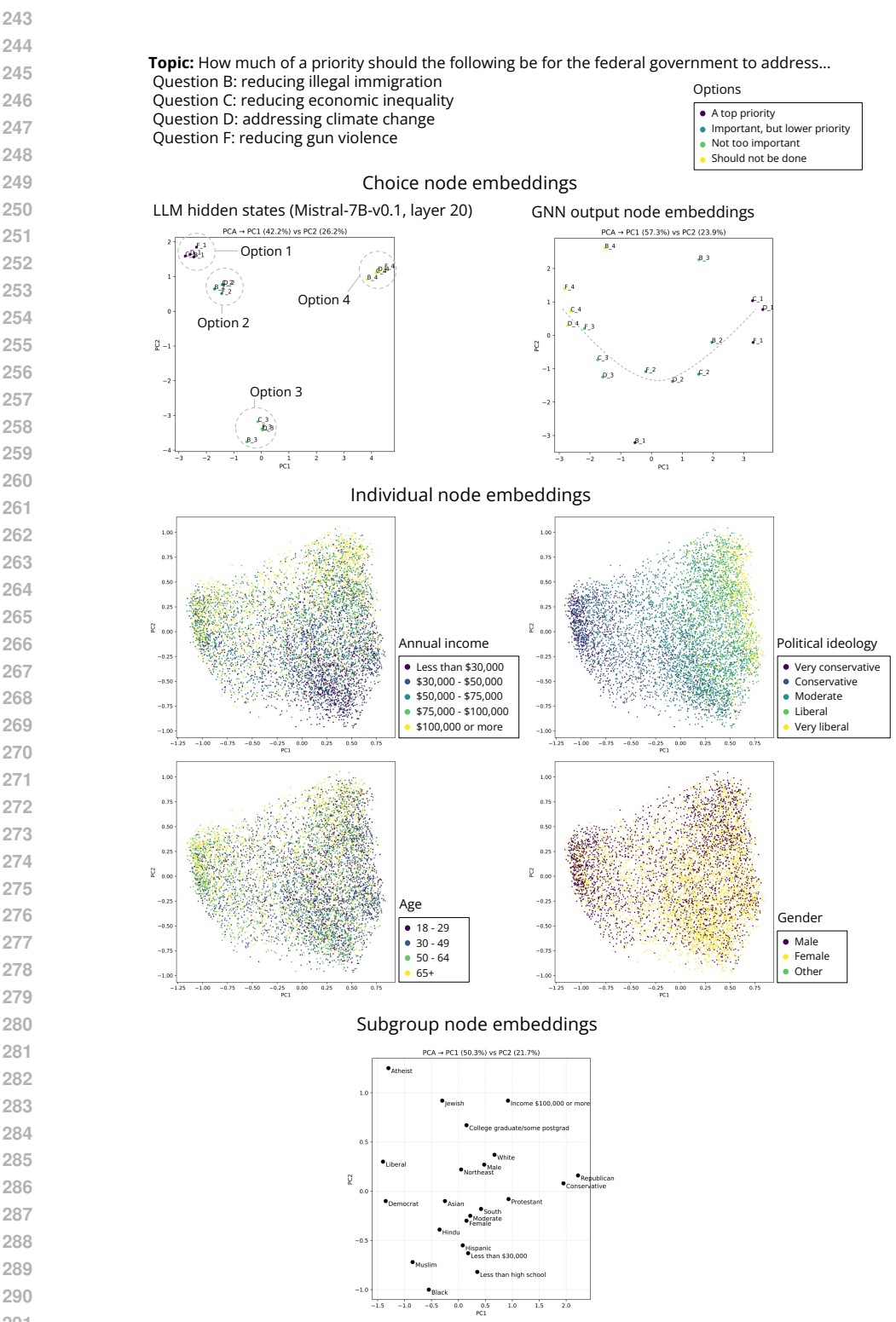

Figure 4: Visualization of LLM hidden states and GNN node embeddings on the first and second components of principal component analysis.

# G ABLATIONS

## G.1 EFFECT OF HIDDEN STATES ACROSS LAYERS

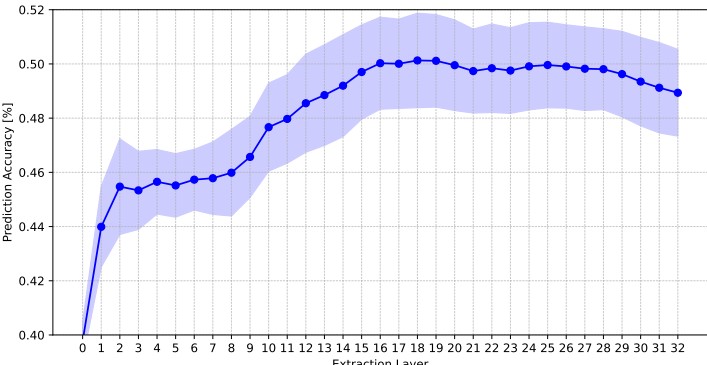

Figure 5: Mean and standard deviation of prediction accuracy on setting 3 (new questions) of OPIN-IONQA dataset when extracting hidden states from different layers of Mistral-7B-v0.1 (Table 3). Layer 0 is the post-embedding activation and layer 32 is the final pre-LM head activation (the model has 32 layers).

Figure 5 shows GEMS accuracy on OPINIONQA dataset with different layers of LLM (Mistral-7B-v0.1) to extract the hidden state from. In practice, we choose the layer that maximizes accuracy on validation questions. Consistent with prior works on probing and interpretability (Kim et al., 2025; Tigges et al., 2023), middle-to-late layers generally provide the most semantically useful and transferable language representations.

## G.2 EFFECT OF THE NUMBER OF LLM–GNN REPRESENTATION PAIRS

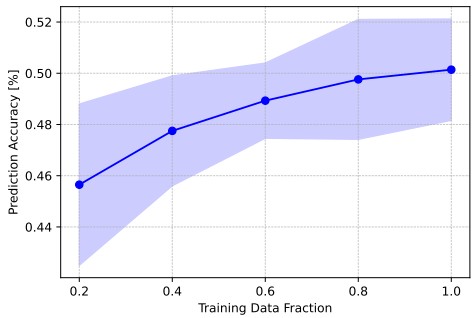

Figure 6: Mean and standard deviation of prediction accuracy on Setting 3 (new questions) of the OPINIONQA dataset using hidden states from layer 18 of Mistral-7B-v0.1. The $x$-axis denotes the fraction of choice nodes in the training graph used to fit the LLM-to-GNN projection in Equation (4). Accuracy improves as more paired examples are used, indicating that sufficient supervision is required to learn a map from LLM hidden states to the GNN output embedding space.

Learning the LLM-to-GNN representation mapping requires paired examples of an LLM hidden state and its corresponding GNN output node embedding. Because this mapping lacks a linguistic prior, performance may degrade sharply when trained on too few pairs. We validate this with an ablation that fits the mapping using only 20%, 40%, 60%, and 80% of the available pairs (fractions taken over choice nodes $\mathcal{C}_{\text{train}}$ in Equation (4)) and evaluate on the new question setting. As shown in Figure 6, reducing the number of pairs leads to a rapid drop in accuracy, showing the sample size sensitivity of the mapping.

## H   PROMPTS TO LLM

We present example prompts for LLM prompting and fine-tuning in the following order:

**Zero-shot prompt:** Provide an individual's features (demographics) in text form, followed by the question. The feature list is determined by available attributes; we primarily use the nine attributes defined in Appendix D. When an individual feature is missing (e.g., age is unknown), we omit it in the prompt rather than explicitly stating its absence (e.g., "Age: unknown").

**Few-shot prompt** (with variable $k$ in-context examples): Provide the individual's features, followed by $k$ prior responses to related questions (see Appendix G for how we select related questions). Append the target question at the end.

**Agentic CoT prompt:** We directly adopt from Park et al. (2024) with minimal modifications. The method consists of two stages. First, the individual's features and prior responses are given to an *expert reflection* module, which produces concise observations about the person's stances. Second, these observations, together with the individual's context, are passed to a prediction module that outputs a an answer in the JSON format.

All examples use synthetic profiles and responses, not real individuals, to protect privacy (Appendix A). For fine-tuning, we apply cross-entropy loss to the single answer token immediately following the input prompt. We note that GPT-OSS (OpenAI, 2025) and Qwen-3 (Yang et al., 2025) use distinct response formats and detail the required tokenization and formatting in Appendix I.

---

**Prompt Example: Zero-shot**

**System**
Respond to the following question by choosing one of the available options, and strictly answering with the option letter (e.g., 'A', 'B', etc.). Do not provide any additional text or explanation.

**User**
Answer the following question as if your personal information is as follows:
Personal identification number: 12345.0
Age: 50-64
Race or ethnicity: White
Gender: Female
Education level: Some college, no degree
Income level: less than $30,000
Region of residence: West
Religion: Nothing in particular
Political party affiliation: Independent
Political ideology: Moderate

Question: Thinking about the nation's economy, how would you rate economic conditions in this country today?
A. Excellent
B. Good
C. Only fair
D. Poor

Answer:

---

---

**Prompt Example: Few-shot** ($k = 2$)

**System**
Respond to the following question by choosing one of the available options, and strictly answering with the option letter (e.g., 'A', 'B', etc.). Do not provide any additional text or explanation.

**User**
Answer the following question as if your personal information is as follows:
Personal identification number: 12345.0
Age: 50-64
Race or ethnicity: White
Gender: Female
Education level: Some college, no degree
Income level: less than $30,000
Region of residence: West
Religion: Nothing in particular
Political party affiliation: Independent
Political ideology: Moderate

Question: How much, if at all, do you think the following proposals would do to reduce economic inequality in the U.S.? Expanding government benefits for the poor
A. A great deal
B. A fair amount
C. Not too much
D. Nothing at all
Answer:

**Assistant**
A. A great deal

**User**
Question: How much, if at all, do you think some people work harder than others contributes to economic inequality in this country?
A. Contributes a great deal
B. Contributes a fair amount
C. Contributes not too much
D. Contributes not at all
Answer:

**Assistant**
A. Contributes a great deal

**User**
Question: Thinking about the nation's economy, how would you rate economic conditions in this country today?
A. Excellent
B. Good
C. Only fair
D. Poor

Answer:

---

---

**Prompt Example: Agentic CoT - 1. Reflection module ($k = 2$)**

**User**
[Participant's information]
Age: 50-64
Race or ethnicity: White
Gender: Female
Education level: Some college, no degree
Income level: less than $30,000
Region of residence: West
Religion: Nothing in particular
Political party affiliation: Independent
Political ideology: Moderate

[Participant's prior responses]
Question: How much, if at all, do you think the following proposals would do to reduce economic inequality in the U.S.? Expanding government benefits for the poor
A. A great deal
B. A fair amount
C. Not too much
D. Nothing at all
Answer: A. A great deal

Question: How much, if at all, do you think some people work harder than others contributes to economic inequality in this country?
A. Contributes a great deal
B. Contributes a fair amount
C. Contributes not too much
D. Contributes not at all
Answer: A. Contributes a great deal

Imagine you are an expert social scientist (with a PhD) taking notes while observing this content. Write observations/reflections about the person's stances about key societal issues. (You should make more than 5 observations and fewer than 20. Choose the number that makes sense given the depth of the content above.)

---

**Prompt Example: Agentic CoT - 1. Reflection module output (truncated)**

**Assistant**
**Observations & Reflections on the Participant's Societal Stances**

1. **Political identity**
- Identifying as an *independent* with a *moderate* ideology indicates a willingness to cross party lines on key issues.
- Such a position often correlates with a preference for incremental, evidence-based reforms rather than ideological extremes.

2. **Attitude towards economic inequality**
...

---

**Prompt Example: Agentic CoT - 2. Prediction module**

**User**
[Participant's information]
Age: 50-64
Race or ethnicity: White
...

[Participant's prior responses]
Question: How much, if at all, do you think the following proposals would do to reduce economic inequality in the U.S.? Expanding government benefits for the poor
...

[Expert social scientist's observations/reflections]
(**Generated observations/reflections from the expert from step 1**)

=====

What you see above is a participant information. Based on the information, I want you to predict the participant's survey responses. All questions are multiple choice where you must guess from one of the options presented. As you answer, I want you to take the following steps:
Step 1) Describe in a few sentences the kind of person that would choose each of the response options. ("Option Interpretation")
Step 2) For each response option, reason about why the Participant might answer with the particular option. ("Option Choice")
Step 3) Write a few sentences reasoning on which of the option best predicts the participant's response ("Reasoning")
Step 4) Predict how the participant will actually respond in the survey. Predict based on the information and your thoughts, but ultimately, DON'T overthink it. Use your system 1 (fast, intuitive) thinking. ("Response")

Here is the question:

=====

Question: Thinking about the nation's economy, how would you rate economic conditions in this country today?
A. Excellent
B. Good
C. Only fair
D. Poor

=====

Output format - output your response in json, where you provide the following:

{"Response": "<your predicted response option letter>"}

---

## I   NOTES ON GPT-OSS AND QWEN3 TRAINING

In this section, we outline the differences in input preprocessing for GPT-OSS (OpenAI, 2025) and Qwen-3 (Yang et al., 2025), which arise from their distinct response formats.

**GPT-OSS.**   GPT-OSS employs the Harmony response format to support advanced context engineering. Each generation typically begins with an analysis channel `<|channel|>analysis`, where the model produces an internal chain-of-thought not exposed to end-users, and concludes with a final channel (`<|start|>assistant<|channel|>final<|message|>`), which contains the user-facing response.

During baseline experiments before fine-tuning, to measure the model's existing predictive capability, we place no constraints on generation: the model is free to produce both analysis and final content, and we parse the output from the final channel to evaluate accuracy.

During fine-tuning, however, we constrain the output to directly generate the answer in the final channel. This step improves predictive accuracy while avoiding social bias that could result from fine-tuning on model-generated chain-of-thoughts, which may yield correct answers through ungrounded reasoning about individuals. In this setup, we append the channel header explicitly to indicate the model that final answer should be generated, and apply next-token prediction loss to the final answer token. An example training input prompt is shown below:

```
<|start|>developer<|message|># Instructions

Respond to the following question by choosing one of the available
options, and strictly answering with the option letter (e.g., 'A', 'B',
etc.). Do not provide any additional text or explanation.

<|end|><|start|>user<|message|>Answer the following question as if your
personal information is as follows:

Personal identification number: 12345.0
Age: 50-64
Race or ethnicity: White
Gender: Female
Education level: Some college, no degree
Income level: less than $30,000
Region of residence: West
Religion: Nothing in particular
Political party affiliation: Independent
Political ideology: Moderate

Question: Would you say the following was a reason or was not a reason
why  there were guns in your household when you were growing up? For
protection

A. Yes, was a reason
B. No, was not a reason

Answer:<|end|><|start|>assistant<|channel|>final<|message|>
```

As shown on the example, the tokenization step involves appending special tokens indicating the final channel. Given the input prompt, the model generates probability distribution over available options in the next token. Cross entropy loss is applied at that token position to fine-tune the model.

**Qwen-3.**   Similarly, Qwen-3 introduces a thinking a mode designed to let the model do more step-by-step reasoning (chain-of-thought) before generating a final answer. During baseline experiments before fine-tuning, we place no constraints on generation and this allows model to perform thinking (wrapped by `<think>...</think>`). During fine-tuning, we constrain the output to directly generate the answer by appending the empty thinking (`<think>\n\n</think>`) explicitly to indicate the model for direct answer generation.

