# OpenReview forum: "Rethinking LLM Human Simulation: When a Graph is What You Need"
_ICLR.cc/2026/Conference — ICLR 2026 Conference Withdrawn Submission_

### Official Review · Reviewer_U1BN · 2025-10-23

**Soundness:** 2
**Presentation:** 2
**Contribution:** 2
**Rating:** 2
**Confidence:** 4

**Summary:**

This paper reframes a classic recommender-style GNN as an alternative to LLMs for discrete choice prediction. While competently executed, it lacks conceptual novelty, overstates its claims about “human simulation,” and provides limited scientific insight.

**Strengths:**

1. The authors clearly reframe discrete-choice human simulation (e.g., predicting survey answers or behavioral decisions) as a link prediction task on a graph, where nodes represent individuals and choices.
2. GEMS achieves comparable or even superior performance to strong LLM baselines (e.g., zero-shot, few-shot, chain-of-thought, fine-tuning) on three human-simulation subtasks.

**Weaknesses:**

1. The presented results are not surprising. It is already well known that large language models perform well on some tasks but not on others. Beating them on structured, discrete-choice problems is expected rather than a breakthrough. The only contribution here appears to be modeling a discrete prediction problem as a network-based one, as graph neural networks for link prediction have already been widely studied in prior work.

2. The use of the term “human simulation” exaggerates the scope of the paper and may mislead readers into thinking that it deals with cognitive modeling, psychology, or game theory, which it does not.

3. The paper’s central empirical claim that graph neural networks can match or surpass large language models is not particularly impressive once it is clear that the evaluated tasks are simple classification problems with small and discrete output spaces.

4. Overall, the work reads more like an engineering benchmark than a piece of scientific research that offers new insights or advances our understanding of the problem.

**Questions:**

1. How can your model be generalized to other tasks related to human simulation?

2. What are the contributions, except for modeling the discrete choice problem as link prediction?

---

> ### Author Response · Authors · 2025-11-21
>
> Dear reviewer U1BN,
>
> We sincerely appreciate your thoughtful comments. We would like to respond to your comment by clustering related points into a single response.
>
> **Contribution of the work and where it is placed in the current literature**
> (Weakness point 1, 3, 4 & Questions point 2)
>
> Here we would like to emphasize that different types of contribution exist.
>
> 1. One type is new methods and models. Our framework is novel in this sense since we have revisited GNNs to tackle simulation problems currently only tackled with LLM-based approaches (Lines 108-116 discuss numerous LLM work to discrete choice settings). Unlike recent work in the simulation space, we haven’t proposed another sophisticated model, for example, adding separate transformer blocks on top of LLM embeddings [1], applying P-tuning [2], or generating synthetic reasoning traces to fine-tune LLMs [3] for discrete choice tasks. Instead we take the opposite path: for a particular but central and widely studied class of simulation problems – discrete choice – we argue that less is enough.
>
> 2. This connects to our second type of contribution: we open up a discussion about the essence of the problem. We are the first to provide a rigorous comparison between LLM-based and graph-based methods for discrete choice. It may be intuitive to some readers (including you) that GNNs can perform competitively with LLMs in discrete settings. Unfortunately and surprisingly, this perspective has been missing in the existing peer-reviewed LLM simulation literature. Prior works, while bringing new training methods as a novelty, do not engage with earlier lines of research on discrete choice modeling and graph-based methods.
>
> 3. As recent and ongoing works increasingly propose the first type of novelty while overlooking simpler and more established approaches to discrete choice modeling [1-7], our contribution is both timely and highly unique: it draws attention back to the core structure of the problem and helps bridge the currently disjoint literatures on LLM-based simulation and discrete choice models. We believe this is why reviewer 6bXm noted “The paper is very well-motivated” with authors demonstrating “very good knowledge of the social simulation literature by covering all the main research progresses in the area.” **If our work is not accepted, this disconnect will only widen, with future papers emphasizing new LLM techniques while continuing to ignore the established discrete choice literature.**
>
> We hope this clarifies the nature of our contribution and how our core message complements and extends the rapidly growing literature on human and social simulation with LLMs.
>
> [1] (ICLR’ 24) Group Preference Optimization
>
> [2] (NAACL’ 24) On the steerability of large language models toward data-driven personas
>
> [3] (EMNLP’ 25) Finetuning LLMs for Human Behavior Prediction in Social Science Experiments
>
> [4] (ICLR’ 25) No Preference Left Behind: Group Distributional Preference Optimization
>
> [5] (ACL’ 25) Language model fine-tuning on scaled survey data for predicting distributions of public opinions
>
> [6] (Nature’ 25) A Foundation Model to Predict and Capture Human Cognition
>
> [7] (NeurIPS’ 25) LLM-Generated Persona Is a Promise with a Catch

---

> > ### Author Response · Authors · 2025-11-21
> >
> > **Use of the term “human simulation”** (Weakness point 2)
> >
> > We appreciate this concern and understand the possible ambiguity of the term “human simulation.” We chose this term deliberately for two reasons: to remain consistent with existing terminology and to match the scope we explicitly define in the paper.
> >
> > **Alignment with existing literature.** We had begun the Introduction by engaging with the terminology used in prior work on discrete-choice settings. Please refer to works cited in Lines 29-34, which tackled discrete choice settings. The cited papers use the following terms: generative agents, survey prediction, human simulation, digital twins, pluralistic alignment / preference optimization, and foundational models. If our focus were restricted to a single domain such as surveys, a narrower term like “survey prediction” would indeed be appropriate. However, our work targets discrete-choice settings more broadly. Among the existing terms, “human simulation” best captures this generality while avoiding exaggeration like “group preference optimization,” which would overstate our contribution.
> >
> > **Clearly delimited scope.** We had explicitly delineated our scope in the third paragraph of the Introduction (Line 43) and further specified concrete instances of simulation in the Problem Definition (Line 126). Throughout, we restricted ourselves to predictive simulation of human choices in discrete settings, and did not claim to model underlying cognitive processes, psychological mechanisms, or strategic game-theoretic reasoning. We will revise the manuscript to clarify our terminology choice and address this potential ambiguity.

---

> > > ### Author Response · Authors · 2025-11-21
> > >
> > > **Generalization capability of GEMS framework** (Questions point 1)
> > >
> > > In the paper, we report experiments on two datasets: (1) OpinionQA public opinion polls and (2) Twin-2K battery covering economic preferences, cognitive biases, and personality traits. To test our framework’s applicability on other simulation domains, we also conducted experiments on a Dunning-Kruger effect replication dataset [8,9], consisting of 20 grammar and logical reasoning questions administered to 3,000 individuals. On this dataset, which evaluates the simulation of individuals’ question-answering patterns, we again observe that the GEMS framework performs comparably to LLM-based methods (Table 1-3). This suggests that our framework is applicable not only to poll- or survey-style responses, but also to human problem-solving patterns.
> > >
> > >
> > > [Table 1] Prediction accuracy, predicting missing responses (setting 1)
> > > | Methods             | LLaMA2-7B     | Mistral-7B     | Qwen3-8B     |
> > > | ------------------- | ------------- | -------------- | ------------ |
> > > |   Random            |               | 20.00                         |
> > > |   Human retest      |               | N.A.                          |
> > > |                     |               |                |              |
> > > |   Zero-shot         | 22.76         | 39.00          | 41.82        |
> > > |   Few-shot (3)      | 26.77         | 43.54          | 46.81        |
> > > |   Few-shot (8)      | 26.34         | 43.59          | 47.60        |
> > > |   Agentic CoT (3)   | 31.01         | 34.41          | 51.18        |
> > > |   Agentic CoT (8)   | 31.68         | 35.45          | 54.71        |
> > > |   SFT               | 56.47         | 56.45          | 56.47        |
> > > |   Few-shot FT (3)   | 56.81         | 56.88          | 56.71        |
> > > |   Few-shot FT (8)   | 57.18         | **57.21**      | 56.94        |
> > > |                     |               |                |              |
> > > | GEMS (RGCN)         |               | 57.68                         |
> > > | GEMS (GAT)          |               | 56.95                         |
> > > | GEMS (SAGE)         |               | **57.89**                     |
> > >
> > > [Table 2] Prediction accuracy, predicting responses of new individuals (setting 2)
> > > | Methods     | LLaMA2-7B | Mistral-7B | Qwen3-8B |
> > > | ----------- | --------- | ---------- | -------- |
> > > | Zero-shot   | 22.44     | 38.83      | 42.06    |
> > > | Agentic CoT | 31.52     | 34.04      | 49.96    |
> > > | SFT         | 56.54     | **56.66**  | 56.50    |
> > > |             |           |            |          |
> > > | GEMS (RGCN) |           | 56.76      |          |
> > > | GEMS (GAT)  |           | 56.70      |          |
> > > | GEMS (SAGE) |           | **57.07**  |          |
> > >
> > > [Table 3] Prediction accuracy, predicting responses for new questions (setting 3)
> > > | Methods         | LLaMA2-7B | Mistral-7B | Qwen3-8B |
> > > | --------------- | --------- | ---------- | -------- |
> > > | Zero-shot       | 19.77     | 43.87      | 44.12    |
> > > | Few-shot (3)    | 23.83     | 49.31      | 52.55    |
> > > | Few-shot (8)    | 23.81     | 44.30      | 52.93    |
> > > | Agentic CoT (3) | 23.56     | 31.18      | 53.54    |
> > > | Agentic CoT (8) | 24.25     | 30.98      | 53.72    |
> > > | SFT             | 33.65     | 48.12      | 43.08    |
> > > | Few-shot FT (3) | 41.18     | 49.28      | **54.13**|
> > > | Few-shot FT (8) | 41.81     | 51.79      | 53.47    |
> > > |                 |           |            |          |
> > > | GEMS (RGCN)     | 47.20     | 47.31      | **52.52**|
> > > | GEMS (GAT)      | 45.02     | 44.57      | 52.01    |
> > > | GEMS (SAGE)     | 46.17     | 47.43      | 52.02    |
> > >
> > > Our Setting 3 where models predict responses for new questions is a form of in-domain generalization, as reviewer 6bXm noted. A cross-domain generalization (e.g., training on political survey data and testing on consumer preference data) is a substantially more challenging and extreme shift in the simulation domain. This is an important and open research direction with existing work suggesting that LLM-based approaches also face difficulties under such shifts [10]. At the same time, in many practical applications it is often sufficient to deploy domain-specialized models (for instance, survey researchers may primarily require high predictive accuracy within survey domains). We will explicitly clarify these points about the scope and limits of our model's generalization capabilities in the Limitations section. We sincerely appreciate your insight on this aspect of simulation tasks.
> > >
> > > [8] “A rational model of the dunning-kruger effect supports insensitivity to evidence in low performers”, Jansen et al., Nature Human Behaviour, 2021.
> > >
> > > [9] “A foundation model to predict and capture human cognition”, Binz et al., Nature, 2025.
> > >
> > > [10] “Take caution in using LLMs as human surrogates: Scylla ex machina”, Gao et al., 2024.

---

> > > > ### Comment · Reviewer_U1BN · 2025-11-26
> > > > **Response to rebuttal**
> > > >
> > > > Thank you for the detailed response.
> > > >
> > > > Although you state that ‘we are the first to provide a rigorous comparison between LLM-based and graph-based methods for discrete choice,’ I do not believe this contribution is sufficient for a top venue like ICLR. Moreover, there are many types of discrete problems, yet this work explores only a few and does not convincingly demonstrate generalizability. Therefore, I will maintain my original score.

---

> > > > > ### Author Response · Authors · 2025-12-03
> > > > >
> > > > > Dear reviewer U1BN,
> > > > >
> > > > > Thank you for your comments. We appreciate your perspective and respect your overall assessment. We would, however, like to clarify the intended scope of our contribution. Our aim is not to cover all possible discrete-choice problems, but to provide a rigorous, controlled comparison between LLM-based and graph-based methods on a set of representative human-simulation tasks. In particular, we evaluate (i) public opinion poll data, (ii) digital-twin survey data, and (iii) tasks capturing individuals’ logical reasoning and grammar patterns. Together, these constitute a meaningful and practically important subset of discrete-choice human simulation settings. We agree that the space of discrete-choice problems is much broader than what we study here, and we would welcome more specific guidance on which additional problem families you view as most critical for establishing generality.

---

### Official Review · Reviewer_FxyC · 2025-10-25

**Soundness:** 3
**Presentation:** 3
**Contribution:** 3
**Rating:** 4
**Confidence:** 4

**Summary:**

The paper introduces GEMS, a graph-based framework that reframes human simulation tasks where individuals choose among discrete options, as a link prediction problem on a heterogeneous graph of individuals, subgroups, and choices. Using relational structure and a GNN (plus an LLM-to-GNN projection when new questions appear), the authors conduct a comprehensive comparison of GEMS against multiple LLM-based baselines, achieving comparable or superior performance in most cases.

**Strengths:**

- The paper provides a comprehensive comparison between the proposed GEMS framework and multiple LLM-based baselines, covering diverse settings and evaluation dimensions.

- It addresses a highly relevant and practical problem and offering an efficient alternative to LLM-heavy approaches.

- The study provides valuable and insightful findings, demonstrating that relational structure and graph-based reasoning can rival or surpass LLMs while being more efficient and interpretable.

**Weaknesses:**

- The related work section is incomplete; it omits relevant studies exploring GNN-based approaches for multi-choice question answering, such as [1]. Including these works would clarify the connection to prior research and more accurately position the paper’s novelty.

- The paper lacks simpler and stronger baselines. Comparing GEMS against a more basic neural network or MLP classifier could better isolate the contribution of the graph structure, while incorporating newer or larger LLMs would help establish upper performance bounds.

- The paper lacks sufficient ablation studies to clarify the specific advantages of the proposed approach. For instance, conducting a user study to assess the claimed interpretability benefits would provide stronger empirical support for those claims.

[1] QA-GNN: Reasoning with Language Models and Knowledge Graphs for Question Answering.

**Questions:**

- The datasets examined in the paper mostly contain a small number of choices (4–6). In real-world scenarios, the number of options can be much larger. How do the authors expect GEMS to perform compared to LLMs in such settings?

- It would be interesting to explore the impact of demonstration selection in the few-shot setting, not only based on question similarity but also by incorporating attributes or other contextual features.

- The paper mentions that LoRA was applied only to the attention query and value matrices. Could the authors clarify the motivation for restricting LoRA adaptation to only these parameters?

---

> ### Author Response · Authors · 2025-11-22
>
> Dear Reviewer FxyC,
>
> We sincerely appreciate your thoughtful comments, which have been very helpful for strengthening the empirical rigor of our work and situating it more clearly within the existing literature. Below, we respond to your questions and concerns in the order they were raised.
>
> ---
>
> **Completeness of Related Work**
>
> We appreciate your concern about the limited engagement with related work in the main paper. In the revised version, we will move the discussion of GNN-based literature from the “Additional Related Work” section currently in the appendix into the main text, taking advantage of the increased page limit from 9 to 10. This include: (1) GNNs for recommender systems, where user-item relationships are modeled on graphs as links; (2) work on text-attributed graphs, where textual information is associated with nodes, as in our setting of predicting responses to new questions; and (3) graph-based question answering methods such as [1]. In addition, to better connect our framework with classical discrete choice modeling (e.g., multinomial logit), we will engage more directly with earlier work; please see references [4-7] in our next response “On the Need for Simpler and Stronger Baselines.” We fully recognize the central role of the Related Work section, especially given our unique contribution of connecting currently disjoint lines of research -- LLM-based discrete-choice human simulation and classical discrete-choice models -- and will accordingly strengthen and expand this section in the revised manuscript.
>
> [1] “QA-GNN: Reasoning with Language Models and Knowledge Graphs for Question Answering”, Yasunaga et al., 2021.

---

> ### Author Response · Authors · 2025-11-22
>
> ## On the Need for Simpler and Stronger Baselines
>
> In response to your suggestion, we implemented two additional baselines: (1) a matrix factorization model and (2) an MLP classifier.
>
> Before describing these results, we would like to mention that we also evaluated GEMS on an additional dataset, Dunning-Kruger effect replication study, which simulates individuals’ problem-solving patterns, to test the applicability of our framework beyond poll- and survey-like datasets. Please refer to our [last response](https://openreview.net/forum?id=tVe3qmrH2h&noteId=HVSGPdewMf) to reviewer U1BN for detailed results.
>
> ---
>
> **Matrix factorization for predicting missing responses, i.e. Imputation (Setting 1)**
>
> For predicting missing responses, a classical discrete choice approach is matrix factorization, which learns embeddings for each individual and choice such that the dot product between an individual and their chosen option is high relative to unchosen options. Concretely, for an individual represented by I1 and a question with three choices represented by {C1,C2,C3}, the model minimizes the cross-entropy between the choice selected by I1 and a softmax distribution over dot products (⟨I1,C1⟩,⟨I1,C2⟩,⟨I1,C3⟩). The main hyperparameter is the embedding dimension for individuals and choices; we sweep over {8, 16, 32, 64, 128}. Matrix factorization can _only_ handle Setting 1, not Settings 2 or 3, since it cannot construct embeddings for new individuals or questions at test time.
>
> **MLP for predicting responses of new individuals (Settings 1 and 2)**
>
> For Settings 1 and 2, we train an MLP classifier for each question. The input is a one-hot encoding of individual features, and the output is a logit distribution over the available choices. During training, the model learns a mapping from individual features to the selected choice, and at test time it predicts responses for new individuals given their features. MLP can therefore handle Settings 1 and 2, but _not_ Setting 3, since it cannot learn a mapping for entirely new questions.
>
> The tables below show the performance of matrix factorization and MLP for Settings 1 (top) and 2 (bottom), compared with existing results.
>
> | Dataset | OpinionQA | Twin-2K | Dunning-Kruger |
> |-|-|-|-|
> | MLP | 50.54 | 62.05 | 56.40 |
> | Matrix Factorization | 52.12 | 62.77 | 56.53 |
> | | | |
> | Few-shot FT Best (3) | 56.31 | 63.91 | 56.88 |
> | Few-shot FT Best (8) | **56.76** | **66.36** | **57.21** |
> | | | |
> | GEMS (RGCN) | 56.89 | 66.36 | 57.68 |
> | GEMS (GAT) | 56.40 | 66.01 | 56.95 |
> | GEMS (SAGE) | **57.00** | **66.62** | **57.89** |
>
> | Dataset | OpinionQA | Twin-2K | Dunning-Kruger |
> |-|-|-|-|
> | MLP | 50.39 | **61.85** | **56.66** |
> | | | |
> | SFT Best   | **50.49** | **61.85** | **56.66** |
> | | | |
> | GEMS (RGCN) | 50.50 | 62.39 | 56.76 |
> | GEMS (GAT) | 50.36 | 62.22 | 56.70 |
> | GEMS (SAGE) | **50.73** | **62.50** | **57.07** |
>
> In Setting 1, where the model must capture the relational structure among discrete choices, **matrix factorization underperforms both GEMS and LLM few-shot fine-tuning on all three datasets, with the MLP classifier performing even worse**. In Setting 2 (new individuals), which relies less on this relational structure, simpler baselines can perform competitively. Combining these two results, we hypothesize that GEMS’ main advantage over classical methods lies in its ability to learn richer relational structure.
>
> We will incorporate these baselines into the main text and discuss how they relate to the classical discrete choice modeling literature [2-5] in the Related Work. Together with our existing comparisons, these additions more clearly isolate the contribution of the graph structure and help position GEMS within both traditional and LLM-based approaches to discrete choice modeling. We believe **these efforts substantially delineate our unique contribution to the field: bridging the currently disjoint literatures on LLM-based simulation and discrete choice models.**
>
> [2] “Discrete choice methods with simulation”, Kenneth E Train, 2009.
>
> [3] “Mixed mnl models for discrete response”, Daniel McFadden and Kenneth Train, 2000.
>
> [4] “Integration of choice and latent variable models”, Ben-Akiva et al., 2002.
>
> [5] “Graph-based methods for discrete choice”, Kiran Tomlinson and Austin R Benson, 2024.

---

> > ### Author Response · Authors · 2025-11-22
> >
> > **Ablation Studies and Empirical Support for Interpretability Claims**
> >
> > We appreciate your suggestions on strengthening the evidence for our interpretability claims. In the paper, we highlighted three main advantages of our GEMS framework: (1) compute efficiency, (2) data privacy and transparency, and (3) interpretability. While (1) is supported by concrete quantitative comparisons (Figure 3) and (2) follows directly from the training setup and data-handling pipeline of GEMS, we acknowledge that (3) is less directly grounded, as interpretability is harder to quantify.
> >
> > Our intention is not to claim that GEMS is fully interpretable in the sense of, for example, linear regression, but rather that it is more interpretable than LLM-based approaches. For instance, Figure 4 (page 24) visualizes node embeddings of choices and individuals, revealing which entities are treated similarly in the representation space and helping us understand the principal dimensions along which they vary -- analysis that is much harder to make within LLMs. We will revise the text in the Advantages section to make this comparative intent explicit and to soften the interpretability claim so that it more accurately reflects what the model currently offers.
> >
> > ---
> >
> > **Scalability to Questions with Many Response Options**
> >
> > While our current datasets involve 2–10 choices per question, our architecture does not fundamentally rely on small choice sets. Graph-based models closely related to GEMS have been widely used in recommender systems, where user-item interactions are sparse and the item catalog can be very large, yet GNN-based recommenders remain competitive. This suggests that our graph-based framework is not inherently constrained by the size of the choice set. Based on this literature, we hypothesize that GEMS would continue to perform well as the number of available choices increases, provided sufficient interaction data is available.
> >
> > At the same time, many discrete-choice human simulation settings we focus on (e.g., people solving multiple-choice questions, respondents selecting among a few options in public opinion polls; Lines 129-137) naturally involve a relatively small number of choices. Systematically benchmarking GEMS and LLM-based methods in regimes with very large choice sets is therefore an open empirical question. We view this as a valuable extension and are keen to explore it once suitable human-behavior datasets with large per-decision choice sets become available.
> >
> > ---
> >
> > **Impact of Demonstration Selection Strategies in Few-Shot Settings**
> >
> > Selection based on question text similarity, which we adopt, is the most common and widely adopted strategy for demonstration selection in few-shot settings [8-10]. We follow this established practice in our experiments. We agree that the demonstration selection strategy can affect LLM prediction accuracy and that exploring alternatives that also incorporate respondent attributes or other contextual signals is an interesting direction. While this is beyond the current scope of our work, we would be excited to adopt and evaluate such improved selection methods as they are developed for human-simulation tasks, and to include them as additional LLM-based baselines in future versions.
> >
> > [8] “Aligning language models to user opinions”, Hwang et al., 2023.
> >
> > [9] “Group preference optimization: Few-shot alignment of large language models”, Zhao et al., 2024.
> >
> > [10] “Few-shot personalization of llms with mis-aligned responses”, Kim and Yang, 2025.

---

> ### Author Response · Authors · 2025-11-22
>
> **Choice in LoRA**
>
> We initially decided to apply low-rank adaptation to query and value matrices following the original paper’s main experiments [11]. We appreciate the reviewer’s feedback on the rigor of our experimental baselines and ran an additional experiment: fine-tuning with larger LoRA rank or full fine-tuning, with results reported in the following table.
>
> **[Table]** Fine-tuning from Mistral-7B-v0.1 with different LoRA ranks or full fine-tuning on the Dunning-Kruger dataset. Settings 1-3 indicate for predicting missing responses, responses of new individuals, and responses for new questions, as in the paper.
>
> |                      |        |                |                |        |        |                |                |
> |:--------------------:|:------:|:--------------:|:--------------:|:------:|:------:|:--------------:|:--------------:|
> | **Setting**              | 1      | 1              | 1              | 2      | 3      | 3              | 3              |
> | **Method**               | SFT    | Few-shot FT, k=3  | Few-shot FT, k=8  | SFT    | SFT    | Few-shot FT, k=3  | Few-shot FT, k=8  |
> | **LoRA, r = 8**          | 56.35  | 56.85          | 56.92          | 56.54  | 41.28  | 46.34          | 40.48          |
> | **LoRA, r = 256 (new)**  | 56.35  | 56.61          | 57.38          | 56.68  | 42.75  | 46.27          | 42.33          |
> | **Full fine-tuning (new)** | 56.35 | 56.68          | 57.36          | 56.58  | 42.78  | 46.25          | 41.85          |
>
> Across all three settings and different numbers of trainable parameters, **we do not observe performance gains from more intensive LLM fine-tuning compared to the results reported in the main paper**. This suggests that our conclusions are robust to the choice of fine-tuning capacity. It also strengths our hypothesis that accurately predicting individuals’ responses hinges on capturing relational structure -- where GNN-based models perform comparably or better to LLMs -- rather than simply increasing the model’s trainable capacity.
>
> [11] “LoRA: Low-Rank Adaptation of Large Language Models”, Hu et al., 2021.

---

> > ### Comment · Reviewer_FxyC · 2025-11-25
> > **Reviewer Response**
> >
> > Thank you very much for your thoughtful response, I truly appreciate the effort you put into addressing my questions and concerns. That said, some of my concerns still remain unresolved:
> >
> > - The issue with the related work is more substantial than simply moving a section or adding a few references. The paper I shared was only one example within a broader family of prior work that has already explored applying GNNs to multiple-choice question answering, which is the main contribution of this paper. This significantly affects the perceived novelty of the work.
> >
> > - Even assuming a fair comparison between baselines, with similar levels of tuning and training effort, the performance gap between the proposed method and the simple baseline remains quite small. This raises questions about the practical impact of the GNN-based approach.
> >
> > - My question about LoRA was primarily about which parameters LoRA was applied to, not the rank. Depending on where LoRA is attached, performance can vary widely.
> >
> > - I still believe that the paper would benefit meaningfully from the suggested ablation studies.

---

> > > ### Author Response · Authors · 2025-12-03
> > >
> > > Dear reviewer FxyC,
> > >
> > > Thank you for the detailed and thoughtful feedback. We respond to your points (1) - (4) in turn.
> > >
> > > **(1) Scope and novelty vs prior GNN - MCQ work**
> > >
> > > We apologize for the confusion here; this is a misunderstanding we did not prevent in the current draft. While both discrete-choice human simulation and CommonsenseQA-style benchmarks (as in QA-GNN paper) superficially appear as multiple-choice question-answering, they differ in a fundamental way: in CommonsenseQA there is a single correct answer and the remaining options are incorrect, whereas in discrete-choice human simulation all options are in principle plausible and the task is to model which option a given individual is more or less likely to choose, given their preferences, prior responses, and background. This setting is therefore closer to personalized recommendation or preference modeling than to standard QA, where the notion of “correctness” is logical or factual, rather than preference-based.
> > >
> > > We fully acknowledge that prior work has already applied GNNs to multiple-choice QA, and we appreciate you pointing out this broader line of research. We do not claim novelty for “GNNs on MCQ data” per se, but rather to (i) adapt graph-based methods to discrete-choice human simulation tasks, and (ii) provide a careful, controlled comparison between such graph-based models and LLM-based methods in this setting. We agree that the paper should more clearly situate itself with respect to the GNN - MCQ literature, and will explicitly articulate both connections and key differences in problem formulation and evaluation.
> > >
> > > **(2) Magnitude and interpretation of performance gains**
> > >
> > > We agree that the Setting 2 result is informative: when individual features are observed and the set of questions is fixed, a simple MLP performs comparably to GEMS. We view this as clarifying the regime in which graph structure is less critical. At the same time, we would like to emphasize that:
> > >
> > > - In Setting 1 (missing responses / imputation), GEMS substantially outperforms the MLP baseline, and
> > > - The MLP cannot be straightforwardly applied to Setting 3 (generalization to new questions), where GEMS continues to perform well.
> > >
> > > These latter scenarios require modeling the relational structure between individuals and questions, and the ability to generalize across that structure, precisely what GEMS is designed to exploit. We appreciate your suggestion to include these additional baselines; together with the new results, they more clearly illustrate our central claim that GNN-based modeling captures relational structure that is crucial for discrete-choice human simulation.
> > >
> > > **(3) LoRA placement**
> > >
> > > Thank you for clarifying your question about LoRA. As mentioned earlier, in our experiments LoRA is applied to the query and value projection matrices in the self-attention layers, following the configuration used in the main experiments of the original LoRA paper. Full fine-tuning, by contrast, updates all parameters in the LLM, not only the query and value matrices. As shown in the table, full fine-tuning does not yield a substantial performance improvement over this LoRA configuration, which suggests that our main conclusions are not driven by an overly restrictive choice of trainable parameters. We will explicitly document the LoRA placement in the main text to avoid ambiguity.
> > >
> > > **(4) User study**
> > >
> > > We appreciate this suggestion and will highlight a prospective user study -- designed to probe the practical advantages of GEMS in real interactive settings. However, within the constraints of the rebuttal period, we focused on adding baselines and more extensive quantitative comparisons, but were not able to carry out a user study in time.

---

### Official Review · Reviewer_zh8M · 2025-10-30

**Soundness:** 3
**Presentation:** 3
**Contribution:** 2
**Rating:** 2
**Confidence:** 5

**Summary:**

The paper presents GEMS, a framework for modeling human choices using GNNs, as an alternative to the use of LLMs. The case is made that GNNs can be at least as good as LLMs for such modeling with better efficiency and interpretability. GEMS uses relational knowledge between humans and tasks and uses link prediction for predicting the human choices on missing responses, new questions, and new individuals. There is a mechanism to transfer representation from LLMs to GNNs for the case of predicting responses on new questions.

**Strengths:**

The paper develops a nice model for predicting choices using link prediction.

The transfer of information from LLMs to GNNs is done well.

The performance and interpretability of the approach is good.

**Weaknesses:**

It is not unexpected that for many domains LLMs' performance can be surpassed through the use of GNNs or some other machine learning method. Thus, I find that the novelty low.

The setup and solutions are sound and along expected lines.

**Questions:**

Why is the approach novel?

---

> ### Author Response · Authors · 2025-11-21
>
> Dear reviewer zh8M,
>
> We sincerely appreciate your thoughtful comments. Here we would like to emphasize that different types of novelty exist.
>
> 1. One type is new methods and models. Our framework is novel in this sense since we have revisited GNNs to tackle simulation problems currently only tackled with LLM-based approaches (Lines 108–116 discuss numerous LLM work to discrete choice settings). Unlike recent work in the simulation space, we haven’t proposed another sophisticated model, for example, adding separate transformer blocks on top of LLM embeddings [1], applying P-tuning [2], or generating synthetic reasoning traces to fine-tune LLMs [3] for discrete choice tasks. Instead we take the opposite path: for a particular but central and widely studied class of simulation problems—discrete choice—we argue that less is enough.
>
> 2. This connects to our second type of novelty: we open up a discussion about the essence of the problem. We are the first to provide a rigorous comparison between LLM-based and graph-based methods for discrete choice. It may be intuitive to some readers (including you) that GNNs can perform competitively with LLMs in discrete settings. Unfortunately and surprisingly, this perspective has been missing in the existing peer-reviewed LLM simulation literature. Prior works, while bringing new training methods as a novelty, do not engage with earlier lines of research on discrete choice modeling and graph-based methods.
>
> 3. As recent and ongoing works increasingly propose the first type of novelty while overlooking simpler and more established approaches to discrete choice modeling [1-7], our contribution is both timely and highly unique: it draws attention back to the core structure of the problem and helps bridge the currently disjoint literatures on LLM-based simulation and discrete choice models. We believe this is why reviewer 6bXm noted “The paper is very well-motivated” with authors demonstrating “very good knowledge of the social simulation literature by covering all the main research progresses in the area.” **If our work is not accepted, this disconnect will only widen, with future papers emphasizing new LLM techniques while continuing to ignore the established discrete choice literature.**
>
> We hope this clarifies the nature of our contribution and how our core message complements and extends the rapidly growing literature on human and social simulation with LLMs.
>
> [1] (ICLR’ 24) Group Preference Optimization
>
> [2] (NAACL’ 24) On the steerability of large language models toward data-driven personas
>
> [3] (EMNLP’ 25) Finetuning LLMs for Human Behavior Prediction in Social Science Experiments
>
> [4] (ICLR’ 25) No Preference Left Behind: Group Distributional Preference Optimization
>
> [5] (ACL’ 25) Language model fine-tuning on scaled survey data for predicting distributions of public opinions
>
> [6] (Nature’ 25) A Foundation Model to Predict and Capture Human Cognition
>
> [7] (NeurIPS’ 25) LLM-Generated Persona Is a Promise with a Catch

---

> > ### Comment · Reviewer_zh8M · 2025-11-27
> > **Not convinced about the novelty of the approach**
> >
> > Thanks to the authors for running many more experiments and the detailed responses.
> >
> > I remain unconvinced of the novelty of the approach.  I just do not think showing that a ML tool is as good or better than LLM in a specific domain is surprising or novel.

---

### Official Review · Reviewer_6bXm · 2025-11-01

**Soundness:** 4
**Presentation:** 4
**Contribution:** 2
**Rating:** 6
**Confidence:** 4

**Summary:**

The authors propose using GNN approaches to model discrete choice human simulation tasks, by formulating the problem as a link prediction problem on a graph of individuals, choices and subgroups. They show that their methods achieve comparable performance to LLM-based approaches.

**Strengths:**

The paper is very well-motivated; The authors demonstrate a very good knowledge of the social simulation literature by covering all the main research progresses in the area. The three evaluation settings make sense. The writing is very clear and easy to follow. It’s also very nice that the authors included test-retest baselines.

**Weaknesses:**

The authors do not adequately address the fundamental reasons why LLM-based social simulator are so popular.

1)	The end-users for these models are often researchers with limited computational resources or expertise. LLMs, accessible via APIs and natural language prompting, offer a near-zero barrier to entry. In contrast, the GEMS framework requires data preprocessing, graph construction, model training, and fine-tuning, all posing a significant technical hurdle.

2)	A key advantage of LLMs is their ability to generate natural language outputs. Even though the chain-of-thought is not a genuine cognitive process, these textual explanations are invaluable for social scientists seeking qualitative insights. GEMS is a purely predictive model and thus simply does not have this capability.

3)	While many papers focus on single-step discrete choice settings, this is certainly not the entire LLM-for-social simulation field, as there are many applications that requires natural language output, or multi-turn interactions.

4)	The paper's "new questions" setting is a form of in-domain generalization. The true challenge, which LLMs are better poised to handle, is cross-domain or cross-dataset generalization (e.g., applying a model trained on political surveys to a new dataset on consumer preferences). The GNN's rigid structure and learned embeddings are unlikely to transfer, a critical limitation that is not discussed with sufficient honesty. The GEMS approach is not a general-purpose "human simulator" but a specialized prediction model.

5) The paper's empirical results are compelling, but their strength is contingent on the choice of baselines. The comparison is made against relatively small (7-8B parameters) and now somewhat dated LLMs (LLaMA-2, Mistral-7B-v0.1). To make a truly convincing case, it is essential to compare against a stronger "upper bound," such as a state-of-the-art proprietary model (e.g., via the GPT-5 or Claude 4.5 APIs) or a strong open model. These models exhibit far superior reasoning and in-context learning capabilities, and practitioners would most likely use them as their first choice. Without this comparison, it is unclear if GEMS's performance advantage holds against the models that are actually being deployed for the social simulation tasks.

6) Several claims need revision

a.	Prompt formulations for LLM (Section 5.1) capture at most 1-hop structure and do not naturally express higher-order dependencies… I would like to see some references on this; In general, as universal function approximators, sufficiently large LLMs can theoretically learn complex, higher-order dependencies from data, even if they lack a specific graph-based inductive bias. This claim should be rephrased as a hypothesis about the differing inductive biases rather than a statement of fact about LLM capabilities.

b.	GEMS makes predictions in a computationally simple and interpretable way -> While the dot-product mechanism and embedding space are more inspectable than an LLM's internal states, GEMS is still a deep neural network, which is fundamentally uninterpretable, compared to, say, a decision tree.


7) Clarification questions:

(a) How does GEMS handle scenarios where the number of available options for a question changes between the training and test sets? This is a common practical issue that LLMs handle seamlessly but would likely require architectural changes or retraining for the GNN.

(b) While the appendices contain details, the main paper would benefit from a more explicit description of the train/validation/test splits, particularly for the more complex imputation setting (Setting 1), to ensure the comparison between methods is clearly understood as fair.

**Questions:**

see above

---

> ### Author Response · Authors · 2025-11-21
>
> Dear Reviewer 6bXm,
>
> We sincerely appreciate your thoughtful comments, which were very helpful in situating our work within the existing literature on LLM-based simulation and more extensively validating our empirical findings.
>
> ---
>
> **1. Accessibility and Implementation Complexity for End-Users**
>
> We acknowledge that GEMS requires graph construction and GNN training with hyperparameter tuning, which introduces more configuration knobs than fine-tuning from an LLM checkpoint (while data preprocessing is common to both GEMS and LLM-based methods). To lower this barrier for practitioners without ML training experience, we (1) highly modularize each step of the training pipeline, and (2) plan to integrate GraphGym for easy-to-use, automated exploration of the GNN design space. [https://github.com/snap-stanford/GraphGym]
>
> ---
>
> **2. On the Lack of Natural Language Outputs in GEMS**
>
> Yes, the ability to generate human-readable rationales behind simulated behaviors can be a valuable source of study for social scientists. For instance, recent work has analyzed models’ reasoning traces [1] although open-ended generations’ practical applicability from social scientists’ perspective remains debated [2]. As we had noted in the Introduction (Line 38), GNNs do not offer this type of natural language outputs, and our scope in this paper is on predictive forms of human simulation which many papers focus on as you pointed out.
>
> [1] “Can Large Language Model Agents Simulate Human Trust Behavior?”, Xie et al., 2024
>
> [2] “'Simulacrum of Stories': Examining Large Language Models as Qualitative Research Participants”, Kapania et al., 2024
>
> ---
>
> **3. Scope Beyond Single-Step Discrete Choice and Multi-Turn Interactions**
>
> Also related to the previous point, there is a growing body of work on human simulation that leverages natural language outputs and multi-turn interactions, with SOTOPIA being one representative example. To better situate our work within this rapidly evolving literature, we will clearly discuss these lines of research in the updated manuscript Related work section.
>
> However, **we emphasize that single-step discrete-choice settings remain a central and widely studied paradigm in LLM-based social simulation**, as had been outlined in our Related Work (works cited in Lines 108-116 consider single-step discrete choice settings with LLMs). Below are several examples of such single-step discrete-choice settings, although the terminology adopted by authors varies across papers (e.g., opinion prediction, preference optimization, foundation models):
>
> (ICML’ 23) Whose Opinions Do Language Models Reflect
>
> (ICLR’ 24) Group Preference Optimization
>
> (EMNLP’ 24) Virtual Personas for Language Models via an Anthology of Backstories
>
> (ICLR’ 25) No Preference Left Behind: Group Distributional Preference Optimization
>
> (ACL’ 25) Language model fine-tuning on scaled survey data for predicting distributions of public opinions
>
> (Nature’ 25) A Foundation Model to Predict and Capture Human Cognition
>
> (NeurIPS’ 25) LLM-Generated Persona Is a Promise with a Catch
>
> (EMNLP’ 25) Finetuning LLMs for Human Behavior Prediction in Social Science Experiments
>
> Given the importance and continued growth of this line of work, we do not view our focus as narrowly scoped, nor as placing LLMs in an artificial or atypical evaluation setting. Instead, our study directly targets a mainstream case in the social simulation literature – one where increasingly sophisticated prompting and fine-tuning methods have been proposed for LLMs – and we see our contribution as a timely addition to this area.

---

> ### Author Response · Authors · 2025-11-21
>
> **4. In-Domain vs. Cross-Domain Generalization of GEMS**
>
> We acknowledge that GNNs may exhibit different generalization capability than LLMs, and that cross-domain generalization -- for example, training on political surveys and testing on behavioral games -- constitutes a much more extreme shift. This is an important and exciting research direction. At the same time, we would like to mention that (1) in many applications, domain-specialized models are sufficient (e.g., for survey researchers, a model with high survey-response prediction accuracy within surveys may already be very useful), and (2) LLM-based approaches face similar cross-domain challenges, even after incorporating more diverse simulation settings into training data [2]. We will make these points explicit in the Limitations section to clearly delineate the scope and limits of our framework in terms of generalization. We sincerely appreciate the reviewer’s insight on this aspect of human simulation.
>
> Additionally, we conduct new experiments to further test the generalization capability of GEMS. Previously, we tested new questions, but now we test entirely new survey waves, conducted with a new panel of respondents, on a new topic and time period. The OpinionQA public opinion polls dataset consists of 14 distinct survey waves covering different survey topics. Following prior work [3], we study topic-level generalization by splitting at the wave-level: we order waves chronologically and use the first 9 waves for training, the next 2 for validation, and the final 3 for testing. Training follows the same procedure as in the main paper (Lines 397-404). Table 1 reports the performance of the GEMS framework and LLM few-shot fine-tuning under this setting:
>
> **[Table 1]** Performance on the OpinionQA dataset Setting 3 (predicting for new questions), testing survey topic-level generalization
>
> | Methods     | k | LLaMA-2-7B       | Mistral-7B-v0.1 | Qwen3-8B        |
> |-------------|---|------------------|-----------------|-----------------|
> | Agentic CoT | 3 | 32.87            | 40.33           | 43.60           |
> | Agentic CoT | 8 | 29.30            | 41.18           | 45.01           |
> | Few-shot FT | 3 | 47.72            | 49.50           | 48.89           |
> | Few-shot FT | 8 | 47.86            | **49.54**       | 49.25           |
> |             |   |                  |                 |                 |
> | GEMS (RGCN) |   | 46.23            | **48.08**       | 47.81           |
> | GEMS (GAT)  |   | 45.79            | 46.96           | 47.03           |
> | GEMS (SAGE) |   | 46.04            | 47.65           | 47.76           |
>
> **In this topic-level generalization setting, GEMS remains comparable to LLM-based methods.** Both GEMS and LLM approaches perform slightly worse than in Table 3 of the main paper, where questions are split randomly rather than at the wave level, reflecting the added difficulty of wave-level generalization.
>
> [3] “Language Model Fine-Tuning on Scaled Survey Data for Predicting Distributions of Public Opinions”, Suh et al., 2025.
>
> [4] “Take caution in using LLMs as human surrogates: Scylla ex machina”, Gao et al., 2024.

---

> ### Author Response · Authors · 2025-11-21
>
> **5. Choice of LLM Baselines and Comparison to Stronger Models**
>
> We appreciate the reviewer’s feedback on the rigor of our experimental baselines and ran two additional sets of experiments:
>
> (1) prompting a state-of-the-art proprietary model (GPT-5 family) and a stronger open-weight model (Gemma-3-27B), with results reported in Table 2.
>
> (2) fine-tuning with larger LoRA rank or full fine-tuning, with results reported in Table 3.
>
> Across these experiments, **we find that GEMS continues to match or exceed the performance of these stronger LLM baselines**. We would like to mention that we also evaluated GEMS on an additional dataset from the Dunning-Kruger effect replication study, which simulates individuals’ problem-solving patterns, to test the applicability of our framework beyond poll- and survey-like datasets. Please refer to our [last response to reviewer U1BN](https://openreview.net/forum?id=tVe3qmrH2h&noteId=HVSGPdewMf) for results.
>
> **[Table 2]** Performance of stronger LLMs on the Twin-2K dataset.
>
> _Top: predicting missing responses (Setting 1). GEMS' performance in this setting is 66.62 (Table 1 in paper)._
>
> | Methods     | k | LLaMA-2-7B       | Mistral-7B-v0.1 | Qwen3-8B        | Gemma-3-27B (new)  | GPT-5-mini (new) |
> |-------------|---|------------------|-----------------|-----------------|-----------------------|------------------|
> | Agentic CoT | 3 | 33.13            | 50.14           | 57.89           |   58.56               | 59.85            |
> | Agentic CoT | 8 | Context Limit    | 48.76           | 60.20           |   60.20               | 60.51            |
> | Few-shot FT | 3 | 63.51            | 63.91           | 62.61           |   -                   |  -               |
> | Few-shot FT | 8 | 65.86            | **66.36**           | 65.27           |   -                   |  -               |
>
> _Bottom: new questions (Setting 3). GEMS' performance in this setting is 60.37 (Table 3 in paper)._
> | Methods     | k | LLaMA-2-7B       | Mistral-7B-v0.1 | Qwen3-8B        | Gemma-3-27B (new)  | GPT-5-mini (new) |
> |-------------|---|------------------|-----------------|-----------------|-----------------------|------------------|
> | Agentic CoT | 3 | 32.16            | 49.67           | 56.18           |  56.34                | 56.83            |
> | Agentic CoT | 8 | Context Limit    | 48.24           | 58.08           |  59.55                | 60.49            |
> | Few-shot FT | 3 | 58.07            | 59.86           | 59.99           |   -                   |  -               |
> | Few-shot FT | 8 | 59.87            | **60.84**           | 60.48           |   -                   |  -               |
>
> **[Table 3]** Fine-tuning from Mistral-7B-v0.1 with different LoRA ranks or full fine-tuning on the Dunning-Kruger dataset. Settings 1-3 indicate predicting missing responses, responses of new individuals, and responses for new questions, respectively, as in the paper. **Increasing the LoRA rank or doing full fine-tuning do not change the model's performance compared to our default in the paper**, LoRA with rank 8.
>
> |                      |        |                |                |        |        |                |                |
> |:--------------------:|:------:|:--------------:|:--------------:|:------:|:------:|:--------------:|:--------------:|
> | **Setting**              | 1      | 1              | 1              | 2      | 3      | 3              | 3              |
> | **Method**               | SFT    | Few-shot FT, k=3  | Few-shot FT, k=8  | SFT    | SFT    | Few-shot FT, k=3  | Few-shot FT, k=8  |
> | **LoRA, r = 8**          | 56.35  | 56.85          | 56.92          | 56.54  | 41.28  | 46.34          | 40.48          |
> | **LoRA, r = 256 (new)**  | 56.35  | 56.61          | 57.38          | 56.68  | 42.75  | 46.27          | 42.33          |
> | **Full fine-tuning (new)** | 56.35 | 56.68          | 57.36          | 56.58  | 42.78  | 46.25          | 41.85          |

---

> > ### Author Response · Authors · 2025-11-21
> >
> > **6(a). Clarifying Our Claim About Prompt Structure and Higher-Order Dependencies**
> >
> > We clarify that “Prompt formulations for LLM …” is not a claim about the expressive capacity of LLMs, but about the structure of the input prompts we pass to LLMs. This is why the sentence begins with “prompt formulations,” not “LLMs.” The prompts used in ours and typically in the discrete choice simulation settings (Appendix H, “Prompts to LLM”) take the form:
> >
> > —
> >
> > Answer the following question as if your personal information is as follows:
> >
> > Age: 50-64
> >
> > Race or ethnicity: White
> >
> > … (list of individual features) …
> >
> > Question: How much, if at all, do you think the following proposals would …
> >
> > Answer: A. A great deal
> >
> > … (list of prior responses) …
> >
> > —
> >
> > This prompt is essentially a verbalization of a 1-hop (depth-1) computational tree with the individual node as the root node. Comparing this prompt to Figure 1, the upper part of the prompt verbalizes the individual’s membership edges, while the lower part verbalizes that individual’s response edges. Sufficiently large LLMs could learn higher-order dependencies from hundreds of thousands of such 1-hop verbalizations. Our point, however, is about inductive bias: GNNs directly aggregate information over neighborhoods within a single forward pass, making them more naturally suited to exploit the higher-order relational structure rather than LLM fine-tuning which sees a large number of 1-hop information.
> >
> > ---
> >
> > **6(b). Clarifying the Interpretability Claim for GEMS**
> >
> > We sincerely appreciate this point and agree that our original words (Line 449: “GEMS makes predictions in a computationally simple … In contrast, it is less direct …”) may overstate the level of interpretability. Our intention was not to argue that GEMS is fully interpretable in the sense of, for example, a decision tree, but rather that it is more interpretable than LLMs, as you noted. We will revise the sentence around Line 449 to make this comparative intent explicit and soften the claim so that it more accurately reflects the model’s actual interpretability.
> >
> > ---
> >
> > **7(a). Clarifying Handling Varying Numbers of Response Options**
> >
> > GEMS is able to handle varying numbers of options for questions, including new numbers of options that appear at test time, and this does not require architectural changes or retraining. Specifically, GEMS first predicts a score between the individual and each question option, then applies a softmax to the scores to produce a probability distribution over options. Thus, GEMS can handle a question at test time with a new number of options by predicting pairwise scores per option then applying the softmax. In fact, the number of options per question frequently changes in the OpinionQA dataset from our experiments, which we will clarify. We tried to show this via Figure 1 (where, in the graph, one question has 3 options and another question has only 2 options) but also noticed that in Line 192 (Choice nodes are structured as a disjoint union…) we haven’t clearly mentioned that number of choices can be an arbitrary number. We appreciate your comment and will make this point more explicit in the revised version.
> >
> > ---
> >
> > **7(b). Clarifying Train/Validation/Test Splits**
> >
> > We agree that a visualization of the dataset splits would be especially helpful for understanding the graph training setup! In the updated version, we will augment the current textual description of train / validation / test splits (Line 319-330 for Setting 1, Line 356-361 for Setting 2, Line 397-404 for Setting 3) with a visualization in Appendix figures.

---

> > > ### Comment · Reviewer_6bXm · 2025-11-24
> > >
> > > Thank you for the detailed response and the additional experiments.
> > >
> > > On (1):
> > > I appreciate the plan to release modular code. However, this remains a distinct limitation: a social scientist can call an LLM API on a laptop CPU in seconds, whereas GEMS requires a training pipeline and likely a GPU. I am not asking you to "solve" this - I accept the trade-off between efficiency/transparency and ease-of-use. Though I think this is a structural barrier that should be acknowledged clearly in the final version.
> > >
> > > On (4):
> > > Regarding generalization, I reviewed your response to Reviewer FxyC. The fact that a simple MLP matches GEMS in Setting 2 (New Individuals) is revealing. It suggests that for new individuals, the graph structure isn't providing much lift over standard supervised classification. Thus, my concern remains that the method is most effective in "imputation" style tasks (Setting 1) where the graph structure is static, rather than generalization to new entities.
> > >
> > > On (5):
> > > Thank you for running the extra baselines, but "GPT-5-mini" and "Gemma-3-27B" do not fully address my request for an "upper bound" comparison. In practice, a researcher attempting to simulate human behavior will likely use the most capable model available (e.g., the full GPT-5 or Claude 4.5). No need to run this if you don't have this result already - just something to think about.

---

> > > > ### Author Response · Authors · 2025-12-03
> > > >
> > > > Dear reviewer 6bXm,
> > > >
> > > > Thank you again for your thoughtful follow-up. We address points (1), (4), and (5) in turn.
> > > >
> > > > (1) We agree that the need for a training pipeline and compute might be a barrier for adoption, and we will make this explicit in the final version. At the same time, we would like to note that the most competitive prompting baseline in our study (Agentic CoT with GPT-5-mini) is also non-trivial to deploy at scale. Because each prediction requires a long chain-of-thought trace, the API cost for a single dataset run was on the order of \$30 - 60, whereas training GEMS on the same dataset required less than an hour on a single GPU (a dollar at current cloud prices). Moreover, achieving the best LLM-based performance in our experiments still required fine-tuning, which (as shown in main paper Figure 3) uses roughly two orders of magnitude more compute than training GEMS. Thus, especially for large-scale or repeated simulations, the comparison is not simply ‘an cost-efficient LLM API on a laptop versus GNN on GPUs’. Nevertheless, we fully agree that the requirement of ML understanding is a barrier for many social scientists, and we will state this as a limitation.
> > > >
> > > > (4) We agree that the Setting 2 result is informative: with individual features and a fixed set of questions, a simple MLP performs comparably to GEMS. We see this as delineating the regime in which graph structure is less critical. In contrast:
> > > >
> > > > - GEMS substantially outperforms the MLP in Setting 1 (missing responses, i.e. imputation), and
> > > > - The MLP cannot be applied to Setting 3 (generalization to new questions) or to our new “new-wave” generalization test, where GEMS continues to perform well.
> > > >
> > > > These latter scenarios require reasoning over the relational structure between individuals and questions, which GEMS is designed to exploit, while a simple MLP cannot represent them in its current form. Importantly, they are also closer to the notion of generalizability in our initial discussion -- generalization scenarios to new questions, waves, and domains -- than the Setting 2.
> > > >
> > > > (5) Upper-bound LLM comparison
> > > >
> > > > Thank you for the suggestion regarding a stronger “upper bound” using the largest available closed models (e.g., full GPT-5). While we do not currently have results with those models, we did analysis by moving from Qwen3-8B to Gemma-3-27B to GPT-5-mini. As shown in Table 2, the gains from scaling across these models are modest and still remain below both the fine-tuned LLM and GEMS in our main settings. This trend and that GPT-5-mini is a very strong proprietary model (as of November 2025) make us cautiously optimistic that our conclusions would persist even with larger closed models. We will add a short discussion acknowledging this limitation and highlighting your suggestion as an important additional experiment.

---

### Author Response · Authors · 2025-12-04
**Summary of responses**

We thank the reviewers for their thoughtful feedback and the area chairs for overseeing the process. Below, we briefly summarize the rebuttal and discussion period and why our paper merits acceptance.

---

### Contribution

Reviewers agree that the paper is clearly written, technically sound, and tackles a timely question: when are LLMs actually necessary for human simulation, and when can smaller, more structured models suffice? Our contributions are:

- We identify and formalize a broad, practically important class of “LLM human simulation” tasks -- discrete-choice simulations (survey responses, behavioral experiments, logical reasoning and answering patterns) -- as a heterogeneous graph link-prediction problem, with an optional mapping from language representations to graph representations.

- We propose GEMS, a GNN-based framework that: (1) achieves comparable or better accuracy than strong LLM prompting and fine-tuning baselines across all settings and datasets, and (2) uses orders of magnitude fewer parameters and compute while providing transparent per-individual and per-choice embeddings that are easy to inspect.

- Through carefully controlled experiments over diverse simulation domains, **we show that -- contrary to current practice and intuition emphasizing increasingly sophisticated LLM-based prompt engineering and training -- an architecture that explicitly exploits relational structure suffices for discrete-choice simulations.**

This gives the community a principled modeling formulation for a large portion of the human-simulation space that has so far been dominated by LLM-centric approaches, and bridges two largely disjoint literatures: LLM-based simulation and graph-based discrete-choice modeling.

---

### Identified concerns

One reviewer (6bXm) is clearly positive and recommends acceptance, emphasizing the importance of the question, the clarity of the exposition, and the strength of the empirical results. The remaining reviews recognize the paper’s technical soundness, the quality of the experiments, and the relevance of the topic, but raise three main concerns:

- the perceived simplicity / sophistication of the method (zh8M, U1BN)
- the scope of our claims relative to “human simulation” more broadly (U1BN, 6bXm)
- the choice and strength of baselines, including frontier LLMs and simpler non-graph models (FxyC, 6bXm)

---

> ### Author Response · Authors · 2025-12-04
>
> ### Our responses
>
> **Perceived simplicity / sophistication of the method**
>
> Reviewers zh8M and U1BN noted that GEMS is methodologically “simple” and that some of the results might appear expected. We agree this is important to address, as simplicity can obscure our work’s substantive contribution.
>
> In our rebuttal, we clarified that our goal is not to propose a new and complex GNN architecture, but to:
>
> - Reformulate a large swath of LLM-based human simulation work as a graph problem, which is non-trivial and had not been systematically done; and
> - Show that this formulation materially changes the empirical and practical answer to a community-wide question: “Do we need LLMs for these tasks?”
>
> The novelty lies in bridging two previously disconnected literatures (LLM human simulation and graph-based modeling), demonstrating that a well-understood class of models provides a strong, principled, and practical alternative in a space currently dominated by LLM-centric methods while neglecting graph-based modeling.
>
> ---
>
> **Scope of claims relative to “human simulation”**
>
> Reviewers asked us to be precise about what our method can and cannot do, and how we use the term “human simulation.” In our rebuttal, we emphasize that:
>
> - Our claims were deliberately scoped to discrete-choice simulation tasks of the kind that are currently often tackled with sophisticated LLM-based approaches (e.g., survey prediction, single-step behavioral decisions).
>
> - We do not claim that GEMS replaces LLMs for open-ended language generation, complex reasoning over natural language, or multi-turn agentic simulations.
>
> - We adopt “human simulation” terminology from existing work to reflect this broader context, while avoiding overclaims such as “preference optimization” or “foundation model” that would suggest a much wider scope than our experiments support.
>
> We believe this scoped claim is both accurate and impactful: it clearly identifies where graph-based models are a better fit than LLMs -- an aspect largely missing from current LLM-based simulation discourse -- and where LLMs remain uniquely performant.
>
> ---
>
> **Baselines**
>
> Reviewer 6bXm requested comparisons to larger, stronger proprietary models, and reviewer FxyC requested simpler non-graph baselines such as MLPs. Both are valid concerns.
>
> During the rebuttal, we added and analyzed these baselines:
>
> - We compared against stronger LLM baselines (including larger, frontier models under reasonable prompting / fine-tuning setups).
>
> - We introduced and evaluated simpler non-graph baselines, including a MLP that operates directly on individual features when available and a matrix factorization.
>
> The additional experiments show that these baselines do not overturn our central empirical conclusion: when fit directly to human response data in the discrete-choice regime we study, our graph-based framework is a very strong competitor to LLM-based methods, often matching or surpassing them while being substantially more efficient and inspectable.
>
> ---
>
> In summary, the reviews collectively acknowledge the paper’s clarity, technical soundness, and relevance. The main concerns – contribution, claim scope, and baseline strength -- have been directly addressed through positioning and additional experiments. What remains is a simple but powerful message, backed by extensive empirical evidence: for a large and practically important subset of “human simulation” tasks, graph-based models like GEMS offer a principled, efficient, and interpretable alternative to LLM-centric approaches.

---

### Note · Authors · 2026-01-06

**Comment:**

Dear Reviewers and Area Chair,

With due respect, we decided to withdraw this submission.
We sincerely appreciate the time and effort invested by the reviewers and the Area Chair in evaluating our work.
The comments were thoughtful and constructive, and they have substantially helped us improve the quality of our work.

Best,
Authors

**Withdrawal Confirmation:**

I have read and agree with the venue's withdrawal policy on behalf of myself and my co-authors.